# Using Alphafold2 to Predict the Structure of the Gp5/M Dimer of Porcine Respiratory and Reproductive Syndrome Virus

**DOI:** 10.3390/ijms232113209

**Published:** 2022-10-30

**Authors:** Michael Veit, Mohamed Rasheed Gadalla, Minze Zhang

**Affiliations:** 1Institut für Virologie, Veterinärmedizin, Freie Universität Berlin, 14163 Berlin, Germany; 2Department of Virology, Faculty of Veterinary Medicine, Cairo University, Giza 12613, Egypt

**Keywords:** arterivirus, porcine respiratory and reproductive syndrome virus, SARS-CoV-2, membrane protein, Gp5, M, Orf3a, protein structure, alphafold, palmitoylation

## Abstract

Porcine reproductive and respiratory syndrome virus is a positive-stranded RNA virus of the family *Arteriviridae*. The Gp5/M dimer, the major component of the viral envelope, is required for virus budding and is an antibody target. We used alphafold2, an artificial-intelligence-based system, to predict a credible structure of Gp5/M. The short disulfide-linked ectodomains lie flat on the membrane, with the exception of the erected N-terminal helix of Gp5, which contains the antibody epitopes and a hypervariable region with a changing number of carbohydrates. The core of the dimer consists of six curved and tilted transmembrane helices, and three are from each protein. The third transmembrane regions extend into the cytoplasm as amphiphilic helices containing the acylation sites. The endodomains of Gp5 and M are composed of seven β-strands from each protein, which interact via β-strand seven. The area under the membrane forms an open cavity with a positive surface charge. The M and Orf3a proteins of coronaviruses have a similar structure, suggesting that all four proteins are derived from the same ancestral gene. Orf3a, like Gp5/M, is acylated at membrane-proximal cysteines. The role of Gp5/M during virus replication is discussed, in particular the mechanisms of virus budding and models of antibody-dependent virus neutralization.

## 1. Introduction

### 1.1. Structure of Arteriviruses and the Function of Their Membrane Proteins

Porcine reproductive and respiratory syndrome virus (PRRSV), an enveloped plus-strand RNA virus in the Arteriviridae family, is arguably the most relevant viral pathogen in pigs worldwide. PRRSV infection causes abortion and stillbirth in pregnant sows as well as respiratory disease and poor growth performance in piglets [1]. PRRSV is divided into two distinct species, PRRSV-1 and PRRSV-2, previously referred to as “European” and “North American” strains [2]. The early isolates Lelystad virus (PRRSV-1) and VR 2332 (PRRSV-2) serve as the prototype strains for the two new species. Since their discovery, PRRSV strains have spread worldwide and have diversified rapidly by mutation and recombination, resulting in the emergence of highly pathogenic variants in China ([3], related to PRRSV-2) and Eastern Europe ([4], related to PRRSV-1). The prototype strain for the Arteriviridae family is the Equine arteritis virus (EAV), which causes substantial disease in horses [5]. Another member is the Simian hemorrhagic fever virus (SHFV), a virus with zoonotic potential [6]. Due to similarities in the genome organization and replication strategy, Arteriviridae are grouped in the order Nidovirales together with the Coronaviridae.

Arteriviruses are composed of a helical nucleocapsid, comprising the viral RNA wrapped by the nucleocapsid protein N, and seven membrane proteins, the disulfide-linked Gp5/M dimer and the GP2/3/4 complex, the small and hydrophobic E-protein and the ORF5a protein [7,8,9]. From reverse genetics experiments, mainly with EAV, it is known that arterivirus structural proteins are essential for virus replication, but act at different steps of the replication cycle. If the expression of either Gp2 or Gp3 or Gp4 is abrogated, virus-like particles bud from cells, but the particles are not infectious, indicating that cell entry is disturbed in the absence of the minor glycoprotein complex. In contrast, if either Gp5 or M is deleted from the viral genome, no virus particles are released from transfected cells. Thus, Gp5 and M are required for virus budding, which does not exclude the possibility that they may have additional functions during virus entry [10,11]. However, the Gp2/3/4 complex governs cell tropism, at least in the cell culture: when the ectodomains of Gp2/Gp3/Gp4 of EAV and PRRSV are swapped, the cell tropism of the resulting recombinant virus is altered, but not by exchanging the genes encoding Gp5 and M [12,13,14].

PRRSV has a restricted cell tropism, infecting alveolar macrophages, the natural host cell in pigs, and MARC-145 cells, a green monkey kidney epithelial-like cell. Initial contacts between the cell and the virus are supposed to be mediated by the binding of M to heparan sulfate proteoglycans, which are ubiquitous cellular membrane proteins [15]. Sialic acid residues on Gp5 are then recognized by sialoadhesin, also known as siglec 1 or CD169, a lectin present on macrophages [16,17]. Although the elimination of sialoadhesin from the porcine genome had no effect on virus replication in experimentally infected pigs [18], other siglecs that can attach to PRRSV have been identified [19]. Attachment is followed by the clathrin-dependent endocytosis of virus particles and their vesicular transport to early endosomes [20,21]. In the endosome, the Gp2/3/4 complex binds to CD163, a scavenger receptor for hemoglobin clearance [22,23]. CD163 is essential for virus entry, since pigs with deletions in this gene are not susceptible to PRRSV infection [18,24,25]. After the receptor-mediated endocytosis of the virus particles, a drop in endosomal pH, accompanied by the proteolytic activity of Cathepsin E [26,27], activates the membrane fusion machinery of PRRSV. The viral membrane proteins that mediate membrane fusion have not been identified so far for any of the Arteriviruses. Possibly, a complex interplay between minor and major viral membrane proteins determines the cell entry of Arteriviruses [23].

### 1.2. Primary Structure and Modifications of Gp5 and M

M is the most conserved membrane protein of the Arteriviruses. It consists of a short ectodomain, three putative transmembrane regions, and a long hydrophilic cytoplasmic tail. M does not have a signal peptide; only M proteins of a few PRRSV strains contain a consensus sequence for the attachment of N-linked glycans, but it is not used [28]. Gp5, the major glycoprotein of Arteriviruses, exhibits basically the same sequence of hydrophilic and hydrophobic domains. The short ectodomain of Gp5 of PRRSV contains two highly conserved N-glycosylation sites, and depending on the virus strain, additional carbohydrates are attached to a hypervariable region at the N-terminus. The subsequent region is mainly hydrophobic and is assumed to span the membrane three times. The C-terminal part is hydrophilic and located in the cytosol, ending up in the virus interior [28].

Gp5 is targeted to the rough endoplasmic reticulum (ER) by an N-terminal signal peptide, and then it is translocated into the ER lumen through the “translocon”, a hetero-oligomeric complex serving as a channel in the ER membrane. During translocation, asparagine residues in the sequence context N-X-S/T (“sequon”) are modified with carbohydrates and the signal peptide is cleaved. The enzyme complexes for N-glycosylation (oligosaccharyltransferase, OST) and signal peptide cleavage (signal peptidase, SPase) are associated with the translocon to perform their activity once a nascent polypeptide chain becomes accessible. Another co-translational modification occurring in the lumen of the ER is the formation of a disulfide bond between Gp5 and M involving the only cysteines in their extracellular domains [8]. For EAV, it was shown that the heterodimerization of M with Gp5 is required for their transport from the ER to the Golgi apparatus, where they are retained by unidentified signals [29]. The Gp5 and M of both PRRSV-1 and PRRSV-2 are palmitoylated at three and two conserved cysteines, respectively, in close proximity to the transmembrane span. This modification is essential for virus replication, affecting the assembly and budding of virus particles [30]. Except for the pivotal role of the Gp5/M dimer, the molecular mechanism of arterivirus budding is largely unexplored. According to electron microscopy studies, virus particle generation proceeds via the budding of pre-formed nucleocapsids into the lumen of the organelles of the exocytic pathway, presumably the Golgi apparatus. After the budding of the virions, they are transported along the secretory route and are ultimately released by exocytosis [31].

### 1.3. Immune Evasion and Persistence of PRRSV 

After acute infection, PRRSV is often not eliminated entirely but continues to replicate at low levels in lymphoid tissues [32]. In the Gp5 of the PRRSV-2 strain VR 2332, one neutralizing epitope was determined in the ectodomain by pepscan analysis and a phage display [33,34]. This epitope was termed ‘‘epitope B’’ since another epitope (‘‘epitope A’’) was identified further upstream [33]. Epitope B elicits an early and strong, but non-neutralizing antibody response, while epitope B appears to be less immunogenic and induces a neutralizing antibody response only late in infection. It was hypothesized that epitope A might work as a ‘‘decoy’’, which is a non-neutralizing, immunodominant epitope that decreases the reactivity of antibodies against a nearby neutralizing epitope [35]. However, the decoy epitope is situated in (or near) the signal peptide of Gp5, and thus the extent and exact position of the signal peptide cleavage critically determine whether it is present in mature Gp5. Mass spectrometry elucidated that the Gp5 of VR 2332 is cleaved at two sites, but probably only the minor fraction of Gp5 molecules in virus particles retains the decoy epitope [36]. Furthermore, a single amino acid exchange in the M of a PRRSV-2 strain confers resistance to a polyclonal swine antiserum with broadly neutralizing activity [37]. For PRRSV-1 strains, a ‘‘decoy epitope’’ could not be identified, but a neutralizing epitope in the ectodomain of Gp5 was described [38]. Note, however, that other glycoproteins of PRRSV, such as Gp4, also contain neutralizing antibody epitopes [39], and the mechanism of PRRSV persistence is complex and controversially discussed [40,41,42,43].

Antigenic drift, i.e., the accumulation of amino acid substitutions in the antibody epitopes of Gp5/M, is one reason why new viral variants continue to emerge in pig farms and why vaccines usually protect only against infection with homologous strains. An additional mechanism by which PRRSV might evade the humoral immune response is glycan shielding, in which the acquisition of a glycosylation site in Gp5 masks a neutralizing epitope [44,45,46].

### 1.4. Alphafold2: An Artificial-Intelligence-Based Method to Predict Protein Structures De Novo

Despite the pivotal role of Gp5/M in the virus replication and evasion of the immune system, no data on the three-dimensional structure of the complex are available. X-ray crystallography of the dimer is challenging because both proteins are hydrophobic and therefore difficult to purify and crystallize. Reconstruction of their structure using high-resolution cryo-EM is currently not possible due to the pleomorphic nature of the virus particles [47]. This leaves bioinformatics as the only option, but accurate predictions of protein structures based on their amino acid sequence were previously difficult. However, a breakthrough was achieved when scientists from Google’s DeepMind used artificial intelligence methods known as deep learning. Their system, called Alphafold, placed first in 2018 in the biennial competition “Critical Assessment of Structural Predictions” (CASP), where research teams predict structures of proteins whose coordinates have been determined experimentally but not publicly released [48,49]. For CASP14 in 2020, the Alphafold system was significantly changed, which led to an unprecedented performance level, especially in the prediction of novel protein folds. Alphafold2 was able to predict the 3D atomic coordinates of folded proteins within the error margin of experimentally determined structures [50,51,52].

Initially, Alphafold was trained on over 180,000 protein structures present in the public Protein Data Bank (PDB) before 30 April 2018. In order to make a prediction, alphafold2 uses the amino acid sequence of the structure to be determined to query large sequence databases to construct a multiple sequence alignment (MSA). The underlying idea is to determine the protein structure from its evolutionary history, which assumes that amino acids covary when making contacts. Simplified, electrostatic interactions between amino acids determine the folding of a protein. Assuming that a residue mutates from a negatively charged into a positively charged amino acid, the contacting amino acid will be under evolutionary pressure to mutate into a negatively charged residue. Otherwise, the contact will be lost, and the resulting protein may not be able to fold, and hence viral proteins viruses containing this residue would die out. From this coevolutionary analysis, it can be concluded that these two amino acids are spatially very close to each other, although they might be far apart in the amino acid sequence. In addition to the MSA, a residue pairs matrix is generated, which initially contains only the information about the amino acid sequence, since residues that are very close in the sequence are also close in space. Alphafold2 can also use the input sequence to search the PDB for similar sequence parts (“templates”), which might exhibit a similar structure. Representations of the multiple sequence alignment (MSA) and residue pairs and information from putative templates are then passed through two neuronal networks, called “evoformer” and “structure module”, and the latter transforms the abstract two-dimensional representations into 3D coordinates of the protein. The whole data are recycled several times through both neuronal networks to improve the prediction [50,51,53]. 

After CASP14, the system was used to predict the structures of the full proteome of humans and from 20 model organisms, which were made publicly accessible in cooperation with EMBL-EBI in the Alphafold Protein Structure Database (https://alphafold.ebi.ac.uk/, accessed on 15 July 2022) [54,55]. After the recent update in July 2022, the database contains over 200 million protein structure predictions, but viral proteins are currently excluded, pending improved support for polyproteins. To allow researchers to make their own predictions, the full code of alphafold2 was subsequently released for download on GitHub https://github.com/deepmind/alphafold, accessed on 21 September 2022. However, running the program requires a high-speed computer with a high memory capacity and advanced computer skills, such as coding, which are not available to everyone. Therefore, independent solutions based on Google Colaboratory (Colab) were developed. They can be accessed on researchers’ local computers from the internet, but computing is performed on high-speed processors in the Google cloud. The field has developed very dynamically as other algorithms were integrated to shorten the computing time to build the MSA, and it was also observed that the system can be used to successfully predict the structure of protein oligomers [56]. See also Ref. [57] for other recent improvements and developments.

Here, we used a web-based version of alphafold2 to predict a model of the Gp5/M dimer of the prototype strains of PRRSV-1 and PRRSV-2. The confidence scores indicated that the models were highly accurate, and in some parts were similar to experimentally determined structures. We describe the individual domains of the dimer and assign its known features, such as protein modifications and antibody epitopes, to the individual amino acids of the structure. The implications of the structures for the function of Gp5/M during virus replication are then discussed, in particular the putative mechanisms of virus entry and budding, as well as current models of antibody-dependent virus neutralization. 

## 2. Results

### 2.1. Evaluation of the Quality of the Gp5/M Dimer Model

We first predicted the Gp5/M dimers of the PRRSV-2 reference strain VR 2332. As an input for the analysis, we used the complete amino acid sequence of M and Gp5, the latter lacking the sequence of the experimentally determined signal peptide [36]. The multiple sequence alignment (MSA) statistic file showed that about 1200 Gp5 and about 100 M sequences were used for the prediction, with mostly full-length coverage (Appendix A). The number of Gp5 sequences was higher because they are used to type PRSSV strains. The program created five models in the form of PDB files and two data sets to evaluate the quality of the predicted models. One of these credibility scores was the “predicted local distance difference test” (pLDDT), which was on average 79 and 77, respectively, for the best two models. Values between 70 and 90 indicate a high accuracy, where the prediction of the main chain of the protein is reliable. The pLDDT values for the five models were also displayed for each residue of Gp5 and M in a graph, where a score between 0 and 100 was plotted against the position of the corresponding amino acid (Figure 1A). The graph for the best-ranked model 1 showed three peaks for Gp5 and M with pLDDT values of 90, which indicate very high accuracy, and are equivalent to structures determined by experiments. These regions corresponded to the three hydrophobic regions of each protein. The endodomains of Gp5 and M were most accurately predicted by model 2, which exhibited pLDDT scores above 90 in this region. The calculated pLDDT scores can also be displayed on the predicted structure in rainbow colors, ranging from red (high confidence) to blue (low confidence), as shown in Figure 1B for model 1.

Another confidence metric is the “prediction aligned error” (PAE) score, which was used to reveal the uncertainty in the distance prediction between individual protein domains (Figure 1C). It displays the calculated error for the predicted distance of each pair of residues as a 2D plot; both axes indicate the position of the individual amino acids. To read the PAE value for a certain pair of residues, a horizontal line is drawn from the position of the first amino acid on the y-axis and a vertical line from the position of the second amino acid on the x-axis. The color at the intersection of the two lines indicates the uncertainty in the predicted distance of the two amino acids, color coded from blue (0–15 Å) to red (15–30 Å). The regions with a blue color in the left part of the graph of model 1 reflect the peaks in the pLDDT score, indicating that the distances between individual amino acid pairs in the ectodomain and the transmembrane regions were confidently predicted. Likewise, distances between amino acids in the endodomain of Gp5 were also accurate. In contrast, the distances between the TM and the endodomain were less accurately predicted. The expected position error was low for the two amino acids present in the same domain, either the TM or endodomain, but it was higher for residues located in two different domains.

In general, the pLDDT score is better suited to display the credibility of the prediction for individual domains, while the predicted alignment error (PAE) is a better analysis of the credibility of the prediction for distances between domains or chains. Thus, the quality of model 1 is better in the extracellular domains and the transmembrane regions of Gp5 and M, whereas model 2 has higher prediction accuracy in the endodomains. However, the differences seem to be rather minor, especially when the exact localization of each amino acid side chain is not important. If the predicted structures of models 1 and 2 are superimposed, they exhibit the same order of their secondary structural elements, α-helices and β-sheets. The backbones of both models also align very well, and differences between the position of individual parts of the protein backbones are only obvious in the ectodomain of Gp5 and the endodomain (Appendix A). Even model 5, which has a low average pLDDT score of only 65.53, revealed the same folding as the other models (not shown).

Furthermore, generating a Ramachandran plot from the PDB file revealed that 95,1% of the amino acids were in the most favored regions, 4.6% were in the additional allowed region, 0.3% (one residue) were in the generously allowed region and none were in the disallowed region (Figure 1D). Generally, good quality models have over 90% in the most favored region, again confirming the validity of the predicted model [58].

In summary, all the confidence scores for the predicted Gp5/M dimer structure of VR 2332 were very high. All the structural elements, such as α-helices and β-sheets, and their topology were predicted with high reliability. The main parts of the structure, the transmembrane regions and the endodomains, were predicted with even higher accuracy, similar to experimentally determined structures, which allowed us to analyze the interactions between the amino acid side chains.

### 2.2. The Structure of Gp5/M of the PRRSV-2 Reference Strain VR 2332

Figure 2 shows a simplified scheme (Figure 2A), a surface representation of hydrophilic and hydrophobic amino acids (Figure 2B) and two cartoons (Figure 2C,D, the latter rotated by 90°) of model 1 of the Gp5/M dimer. The known protein modifications are highlighted in the cartoon representations, with glycosylated asparagines highlighted as blue spheres and palmitoylated cysteines as orange spheres. Red sticks in the ectodomain indicate the disulfide bond between Gp5 and M. The negatively charged Asp58 in the TM2 of Gp5 is also shown as a red stick. See also the graphical abstract displaying the Gp5/M dimer embedded in a membrane, including the localization of the protein modifications. The assignment of the secondary structural elements to the amino acid sequence is depicted in Appendix A.

The first thing to notice is that the structures of Gp5 and M look very similar. The short N-glycosylated ectodomain of Gp5 is composed of two α-helices. The non-glycosylated N terminus of M consists of a short α-helix containing the cysteine that forms the experimentally established disulfide bond with the cysteine in Gp5. The core of the dimer is composed of six helices, three from Gp5 and three from M. The calculations of hydrophilic and hydrophobic regions revealed that most of the surface residues of the helices were hydrophobic (colored light brown in Figure 2B) and hence represent the membrane-spanning part of the dimer. The six helices do not run straight and parallel through the membrane but are tilted, and helix 1 and helix 3 of both proteins are also kinked or curved. The C-terminal part of transmembrane helix (TM) 3 contains the acylated cysteines, two in M and three in Gp5 (highlighted as orange spheres). The endodomains of Gp5 and M are composed of β-strands, seven from each protein, which interact with β-strand seven. They are connected via unstructured linkers to the respective TM3, giving the structure of the endodomain a basket-like appearance. The surface representation of the Gp5/M dimer also shows that the endodomain has a cavity below the transmembrane part that is open to both sides.

### 2.3. Structure and Diversity of the Ectodomain of Gp5/M of VR 2332 and Its Epitopes

Figure 3A shows a cartoon and Figure 3B is a surface representation of the ectodomain of the Gp5/M dimer. It lies flat on the transmembrane regions, except for the α-helical N-terminus of Gp5, which protrudes with an angle of about 45 degrees at one edge. The distance between the N-terminus of Gp5 and the membrane-spanning region is only 20 Å. Such a rather flat ectodomain is consistent with the smooth and featureless outer surface of the Arterivirus particles observed with electron microscopy [47]. The helix contains one N-glycosylation site near the N-terminus and another site at its end, the latter is conserved in all PRRSV strains (N-glycosylation sites are highlighted as blue sticks in Figure 3A and as blue spheres in Figure 3B). The second helix in Gp5 contains the third conserved N-glycosylation site, and it is connected via a disulfide bond to the helix in M (red sticks in Figure 3A). A large variety of mostly complex-type carbohydrates, some of them with terminal sialic acid moieties, have been identified in Gp5 [59]. Due to the flat surface of the ectodomain, sugars linked to every site are exposed at the molecule`s surface and hence can be recognized by sialoadhesins, which are viral attachment factors.

We then asked where in the 3D structure of Gp5/M are the antibody epitopes located, and whether their amino acid sequence varies between PRRSV-2 strains. To determine which residues of the ectodomain are variable, we downloaded 7736 Orf5 and 290 Orf6 nucleotide sequences of PRRSV-2 strains from the NCBI database and translated them into the corresponding amino acid sequences. From these data, a consensus sequence containing the most abundant amino acid at each position was compiled, and the percent conservation at each position was plotted against the amino acid number (Appendix A). We also generated a web logo representation from the sequences to display which amino acids are present at each position and at which frequency (Figure 3C,D).

The neutralizing epitope in the Gp5 of VR 2332 encompasses residues 5 to 12 (shown as red or cyan spheres in Figure 3B), which are located at the C-terminal part of the N-terminal helix. (Note that the numbering of residues starts with the first amino acid in the mature protein, which corresponds to residue 32 of the full-length protein including the signal peptide). The web logo representation shows that only three residues of the Gp5 epitope (7, 10 and especially 8) were variable between strains, and the others (5, 6, 9, 11 + 12) were almost completely conserved. Many of the conserved epitope residues (red spheres) were exposed on the upper side of the helix, while the variable residues (cyan spheres) were located on its underside. Residue 10 of M was the most variable residue in the ectodomain of M, and could harbor amino acids with large side chains, such as Asn, His, Tyr or rarely Met, as in the case of the M from VR 2332 (Figure 3D). It has been described that substitutions at position 10 provide an escape from broadly neutralizing antisera [37]. Residue 10 in M was located at the molecule’s surface (highlighted as a wheat sphere in Figure 3B) close to the antibody epitope in Gp5, and this suggests that they formed a common conformational epitope.

The remainder of the ectodomain of M was (almost) completely conserved, including Cys9, which formed the disulfide linkage with Gp5 (Figure 3D). The acceptor of the disulfide bond in Gp5, Cys 17, was also completely conserved, as was the central part of Gp5’s ectodomain sequence, including two N-glycosylation sites (Figure 3C). The ectodomain of Gp5 had two hypervariable regions (magenta spheres in Figure 3B). One comprised residues 26–28, which were located in the loop that connected the ectodomain to TM1. The region contained mostly hydrophilic amino acids with large side chains, such as Asn, Asp, Arg and Lys, which were exposed at one edge of the molecule’s surface. No function has been attributed to this region so far. The second hypervariable region encompassed residues 1 to 4 located at the beginning of the N-terminal helix. The region almost exclusively contained Asn and Ser and rarely Thr which, together with the highly conserved Ser at positions 5 and 6, could form a large variety of N-glycosylation motifs NXS/T. Thus, mutations occurring frequently in residues 1 to 4 during virus evolution can either remove or cause the attachment of an additional carbohydrate to Gp5. Some strains even contained the sequence NNSS, which corresponds to two overlapping glycosylation sequons. One or two carbohydrates attached to residues 1 to 4 at the beginning of the α-helix would probably prevent the access of antibodies to the neutralizing epitope (residues 5–12).

It has been proposed that negatively charged heparan sulfate binds to Gp5/M to mediate cell attachment of PRRSV [15]. Therefore, we calculated the electrostatic potential of its ectodomain to identify clusters of positively charged amino acids, which are known as heparan sulfate binding sites [60]. The calculation, however, revealed only a region with a strong negative charge, which was composed of three conserved Asp residues from M and Asp 30, the first amino acid of the TM1 of Gp5 (Figure 3E).

Gp5 is proteolytically processed in acidic, recycling endosomes by Cathepsin E, which might activate the fusion machinery of the virus [26]. Cathepsin E preferentially cleaves between hydrophobic residues, which are abundant at the flat and conserved part of the ectodomain of Gp5; hence, cleavage would remove the variable N-terminal helix.

Finally, to identify additional residues that might be exposed at the surface of the Gp5/M dimer, we used a bioinformatic tool that placed the Gp5/M dimer in a virtual bilayer with a thickness of 30Å, the size of the Golgi membrane. It revealed that hydrophilic residues in the loop connecting TM2 and TM3 were protruding from the membrane, four in M and three in Gp5 (Figure 3F). These residues might also contribute to the interacting capabilities of the Gp5/M dimer during virus entry.

### 2.4. Structure of the Longer Ectodomain of Gp5/M of VR 2332 Containing the Decoy Epitope

The non-neutralizing decoy epitope is present in mature Gp5 of VR 2332 only if signal peptide cleavage occurs upstream of its major cleavage site between residues 31 and 32 [33,34,36]. To determine whether or not cleavage at this site affects the tertiary structure of Gp5/M and where the decoy epitope is located, we created a model using the longer version of the Gp5 of VR 2332 as the input. The resulting structure exhibited essentially the same folding as Gp5 cleaved between 31 and 32, only the N-terminal helix was extended by five amino acids (Appendix A). The decoy epitope Val, Leu, Ala and Asn included the first turn of the N-terminal α-helix and was surface exposed. Its sequence was rather variable between the PRRSV-2 strains, but mostly hydrophobic amino acids were present at these positions (Appendix A). Since the last amino acid of the epitope was an asparagine in the Gp5 of VR 2332 (as well as in many other PRRSV-2 strains), cleavage at residue 26 added the additional glycosylation site NAS to the mature protein. A subfraction of the Gp5 molecules in the VR 2332 particles contained a carbohydrate at this position [36].

We also predicted the structures of the Gp5/M of viruses belonging to four different sublineages of PRRSV-2 [61]. The results, displayed in Appendix A, revealed that the folding of each dimer was consistent with the predicted structure of the Gp5/M of VR 2332. The confidence metrics for the transmembrane region and endodomain (shown as pLDDT displayed on the predicted structures in Appendix A) were good, although slightly different between molecules. The ectodomains of Gp5 and M were helical and each Gp5 contained one carbohydrate attachment site at the beginning and two at the end of the N-terminal helix. However, the orientation of the helices relative to the transmembrane region differed between the Gp5/M proteins. In the Gp5/M from lineage 3.1, the Gp5 helix was upright, as it was in the Gp5/M of VR 2332, and this model had the best pLDDT score. In the model from lineage 3.7, the helices from M and Gp5 were flat, and in the models from lineage 2.2 and 2.6, the Gp5 helices were flat and the M helices were erected. The latter two models had the worst pLDDT scores, and no disulfide linkage between Gp5 and M was predicted. Thus, it appears that alphafold2 correctly predicted the secondary structure of the ectodomain, but not necessarily the spatial arrangement of the individual elements with respect to each other. The orientation of the ectodomain was determined by the angle between the TM1 and α-helix of Gp5; hence, small uncertainties in the prediction of this loop region had a strong impact. Therefore, it cannot be inferred from these models whether the antigenic drift in Gp5 that occurs during virus evolution causes a conformational change in the ectodomain.

### 2.5. Predicted Signal Peptide Cleavage Sites in Gp5 Proteins of PRRSV-2 Strains

To estimate whether the Gp5 of other PRRSV-2 strains might contain a decoy epitope, we used SignalP5 to predict signal peptide cleavage sites for 146 Gp5 from other PRRSV-2 strains. The bioinformatics tool calculates the probability of whether the N-terminus of a protein functions as a signal peptide and which site is preferentially cleaved. For 10 Gp5 proteins, the N terminus was predicted not to function as a signal peptide. From the remaining 136 sequences, 119 (87.5%) were predicted to be cleaved between positions 31 and 32; seven (5.1%) between residues 29 and 30; six (4.4%) between residues 26 and 27; and one Gp5 protein each between 24 and 25, 27 and 28, 28 and 29 and 34 and 35, respectively. The probability for a particular cleavage site was rather low and varied from 17% to 82% for individual Gp5 proteins (Appendix A). For many Gp5 proteins, a second or even third cleavage site was predicted (data not shown).

The removal of a signal peptide requires the presence of small amino acids, Ala, Val, Ser, Cys or Gly, at the −1 and −3 positions of the cleavage site [62,63]. The web logo displaying the frequency of amino acids in the signal peptide of ~5000 Gp5 proteins showed that small amino acids were highly conserved at positions 31 and 29, which explains the preferential cleavage between 31 and 32 (Appendix A). Position 27 also only contained small residues, which together with position 29 created another pair of small residues targeting cleavage to position 29. Moreover, position 26 contained only Val and Ala, and position 24 often contained Cys, which allowed cleavage between 26 and 27. Thus, the signal peptide of the Gp5 proteins of the PRRSV-2 strains contained four positions where the small amino acids were highly conserved, and one position where they were abundant. This creates a large number of putative signal peptide cleavage sites, and the sequence context then likely determines which one is preferentially used. The substitution of amino acids during viral evolution could shift the cleavage of the signal peptide to a different site or allow cleavage at a second or third site in a subfraction of molecules, which in turn could lead to the presence of a decoy epitope in the virus particles.

We also predicted the signal peptide cleavage sites of the Gp5 of the representative members of the three clades of the PRRSV-2 strains and their sublineages [61]. Every Gp5 protein had the main cleavage site between residue 31 and 32, but for 10 of the 15 sequences, a second minor site upstream of the main site was predicted. The alignment of the sequences revealed that the main cleavage site contained a small amino acid at the −1 and at the −3 position. The second sites were also specified by small residues, but these sites contained large hydrophobic residues in the Gp5 sequences with just one cleavage site. Similarly, the Gp5 proteins predicted with the highest probability to be cleaved mainly at a site other than 31/32 contained a large residue at either position -1 or -3, thereby blocking the main cleavage site (Appendix A).

### 2.6. Peculiar Features of the Transmembrane Region of Gp5/M of VR 2332

Viewed from the extracellular side, the transmembrane helices (TMs) traced the circumference of an ellipse with the TM1–3 from M followed by the TM1–3 of Gp5 in clockwise order (Figure 4A). The TM3s were positioned at the main axis vertices at a distance of ~36Å between the N-termini of TM3 and ~47 Å at their cytoplasmic ends. The TM1s and TM2s packed against each other across the elliptical minor axis, and the distance between the cytosolic start of the TM2 of M and Gp5 was ~19Å. The placement of the Gp5/M dimer in a virtual bilayer (Figure 4B) revealed that the TM1 of Gp5 and M did not enter the membrane perpendicularly, but at an angle of about 20 degrees (Figure 4A,B). Each TM1 helix was bent by a completely conserved proline (shown as wheat sticks) in the middle so that from then on, they ran almost parallel to each other and vertically through the membrane. Short loops (3–4 amino acids), which were inside the membrane bilayer, connected TM1 and TM2, and the latter was not curved, but it also ran at an angle through the membrane. Another loop, which was partially exposed at the molecule’s surface (Figure 3F), connected to the TM3s, which were the longest helices and also ran angular through the membrane.

Previously, it was unclear whether helix 2 of the Gp5 spans the membrane because residue 58 in its middle is a negatively charged Asp [8]. The Gp5/M model predicts that the carboxyl group of Asp58 forms an ionic bond with Lys35 in M, which is located in the loop between TM1 and TM2 and also interacts with Thr43 in the TM1 of Gp5. These interactions neutralize the charge of Asp58 and enable its insertion into a lipid bilayer. There are two more electrostatic interactions connecting the transmembrane regions of Gp5 and M. In the outer membrane leaflet, the hydroxyl group of Tyr81 in the TM3 of Gp5 interacts with the main chain in the TM1 of M at position Thr25. The side chain of this Thr interacts with the main chain of the ectodomain of Gp5 at position Leu 16. Finally, Glu34 in the TM1 of Gp5 interacts with the main chain of M at position 75 located at the end of the loop connecting TM2 and TM3 (Figure 4A,B). The amino acids forming these interactions are completely conserved in the Gp5 and M of PRRSV-2 strains (Appendix A) and hence are probably essential to stabilize the unusual orientation of the transmembrane regions within the lipid bilayer. In general, the amino acids in the six helices are very conserved or even invariant. This suggests that they perform a specific function that goes beyond membrane anchoring. More variable are the residues located in loop 2 connecting TM2 with TM3 in both M and Gp5, and some of them are exposed above the membrane.

The placement of the Gp5/M dimer in a virtual bilayer revealed that the membrane-spanning parts of helix 3 were terminated by charged residues, Arg95 in Gp5 and Glu90 in M, and hence the remainder of both helices extended into the cytoplasm (Figure 4B, graphical abstract). The bioinformatics tool Heliquest predicted that these C-terminal parts of helix 3 had an amphipathic character; one part of the helix contained hydrophobic and the other part hydrophilic amino acids (Figure 4C,D). This analysis was supported by the predicted structure: most hydrophobic residues, especially those with large side chains, such as Phe, Trp and Ile, were located on the membrane-near side of the helix, while the charged residues were oriented towards the cytoplasm. The membrane-near part of both helices also contains the conserved palmitoylated cysteine residues (highlighted in orange in Figure 4) [30], and the fatty acids attached to them would increase the hydrophobicity of this region further. Helices with such biophysical properties can be partially inserted into the lipid bilayer to cause membrane curvature, a process that might facilitate the budding of enveloped virus particles [64,65].

### 2.7. The Basic Endodomain of Gp5/M of VR 2332

We used the model ranked 2 with the best pLDDT scores for the endodomain to describe the details of its structure. The endodomains of Gp5 and M exhibited the same secondary structural elements, which were arranged in the same topology (Figure 5A). They were composed of seven β-strands, which were connected by a linker to TM3. In both proteins, lateral interactions between β1, β2, β6 and parts of β7 build one β-sheet, and interactions between β3, β4 and β5 build another β-sheet, and both are arranged in an antiparallel orientation. Gp5 and M interact mainly via the longest β-strand 7, which forms a β-sheet with an antiparallel orientation, but they are twisted against each other. In addition, this β-sheet also contains portions of the β3 of Gp5 and M, where the β3 of Gp5 interacts with the β7 of M and the β3 of M interacts with the β7 of Gp5. In addition to these hydrogen bonds between the β-strands, the amino acid side chain of Glu164 in the middle of β7 in Gp5 forms a salt bridge with Lys163 in β7 in M and with Arg136 at the end of the β5 in M (Figure 5B). All of these residues are highly conserved between virus strains (Appendix A).

A surface representation revealed the most striking feature of the structure: a cavity with openings on both sides, which gives the endodomain a basket-like structure, with handles connecting to the TM region (Figure 5C). The dimension of the endodomain was 42Å in the horizontal direction, i.e., the distance between the C-termini of the TM3 of Gp5 and M, and 33Å in the vertical direction, i.e., the distance between loop 1 and β7 in M. Calculations of the surface electrostatic potential revealed a positively charged molecular surface, especially in the cavity (Figure 5D). This was due to a number of basic amino acids in Gp5 and M, which point their side chains toward the interior of the cavity, narrowing the channel. All of these residues were highly conserved between virus strains (Appendix A). An acidic amino acid, Glu 139 in Gp5, oriented its side chain toward the opening. However, it was not conserved and, in some strains, was replaced by Gly or Asn.

### 2.8. Model of the Gp5/M Dimer of the PRRSV-1 Reference Strain Lelystad

We used the amino acid sequences of the prototype strain Lelystad to predict the structure of the Gp5/M dimer of a PRRSV-1 strain. The resulting model had an average pLDDT score of ~80, similar to values obtained for the Gp5/M of VR 2332. Especially the endodomain was predicted with a very high accuracy (Appendix A). The Ramachandran plot statistics were very similar to the ones obtained for Gp5/M of VR 2332, with 94.6% of the amino acids in the most favored regions, 5.4% in the additional allowed region, and none in the generously allowed or disallowed region. The assignment of the secondary structural elements to the amino acid sequence is depicted in Appendix A.

The Gp5/M of Lelystad and VR 2332 exhibited the same folding, and the backbone atoms of both dimers aligned very well, with an average distance of a 3.7Å root-mean-square deviation (RMSD) (Appendix A). The Lelystad Gp5/M dimer exhibited the characteristic features of the PRRSV-2 dimer: a disulfide linkage between M and Gp5 connected the ectodomains; proline residues caused a kink in the TM1 of Gp5 and M; and a salt bridge between Lys34 in the TM1 of Gp5 and Asp59 in the TM2 of M connected the TMs of both subunits. Placing the Gp5/M dimer into a virtual bilayer revealed that the residues located in the loop connecting TM2 and TM3 were exposed above the membrane (Appendix A). The acylated cysteines, three in M and three in Gp5, were located near the C-terminus of TM3, which extended into the cytoplasm (Figure 6A). Using heliquest to predict the biophysical properties of this part revealed that they were amphiphilic, and the structure showed that hydrophobic residues, including four of the five palmitoylation sites, were located on the side of the helix that was close to the membrane (Appendix A). A calculation of the electrostatic surface potential revealed an open cavity below the transmembrane region with a highly basic surface due to the presence of positively charged amino acids in both Gp5 and M pointing into the interior of the cavity (Figure 6A,B). The endodomain exhibited the same topology of β-strands, and Gp5 and M interacted mainly via β7 and via three electrostatic interactions (Figure 6C).

One neutralizing antibody epitope encompasses residues 4 -18, but no decoy epitope has been identified in the Gp5 of Lelystad [38,66]. The epitope was thus longer than that in PRRSV-2, but only the amino acids surrounding the N-terminal helix were exposed at the surface of the dimer (Figure 6D). An analysis of ~1000 PRRSV-1 Gp5 sequences revealed that the epitope, except residue 4, was conserved between strains. The two N-glycosylation sites, N12 and N19, and cysteine 16 forming the disulfide linkage to M were also conserved. More variable were residues 1 to 4, which often contained Asn and Ser, which together with the conserved Ser and Thr residues at positions 5 and 6 could form different glycosylation motifs. The hydrophilic amino acids exposed at the C-terminal end of the ectodomain (residues 21, 24–29) were also mostly variable. In contrast to the PRRSV-2 strains, the ectodomain of M was more variable between the PRRSV-1 strains than the ectodomain of Gp5 (Figure 6E,F and Appendix A).

We previously showed with mass spectrometry that the signal peptide of the Gp5 of a variant of the Lelystad strain is cleaved between positions 34 and 35 [67]. However, SignalP5 predicts for 180 Gp5 sequences that the majority of them (135, 74.4%) are cleaved at another site, between residues 32 and 33. A total of 30 Gp5 proteins (16.6%) were predicted to be cleaved between 34 and 35; 14 (7.7%) between positions 30 and 31; and 1 between residues 36 and 37. The mean of the probability scores was low (54 + −12%), exhibiting also a large difference between the maxima (85%) and minima (12%) (Appendix A). For many Gp5 proteins with a low probability to be cleaved at a particular site, a second or third site was also predicted in the graphical summary of the results (not shown). The web logo representation of the ~2000 Gp5 sequences revealed conserved small residues at positions 34, 32, 30 and 28, which in principle allows cleavage at the predicted sites (Appendix A).

### 2.9. Model of Monomeric Gp5 Suggests Conformational Changes Occurring upon Dimerization

The membrane projection of Gp5/M showed that the two conserved glycosylation sites were very close to the membrane (Figure 3F and Appendix A). This is an unusual position since it might prevent the attachment of sugar chains by steric hindrance, i.e., the glycosylation sequon is not accessible by the membrane-bound oligosaccharyltransferase (OST) [68,69]. N-glycosylation occurs co-translationally, most likely on the monomer, and therefore we predicted the structures of the monomeric Gp5 and M of VR 2332 and compared them with the structure of the corresponding dimers. The calculated pLDDT scores of the monomeric Gp5 were only slightly lower compared with the Gp5/M dimer, between 60 and 80, even up to 90 in the TM3 of Gp5 (Figure 7A,B). An exception is the structure of the endodomain of the M protein, which exhibited pLDDT scores between 40 and 60 (Appendix A).

The transmembrane regions of Gp5 aligned well with the corresponding regions in the dimer, but a deviation was visible in the ectodomain. The N-terminal helix was shorter and more erect in the monomer. The two conserved glycosylation sites, N20 and N13, and the variable site N2, were 10Å, 13Å and 12Å, respectively, farther from the membrane in the monomer (Figure 7C). We assume that they were drawn closer to the membrane by the formation of the disulfide linkage with the cysteine in the short ectodomain of M. This conformational change, which also affects parts of the transmembrane region and the endodomain, could also make Gp5/M competent for transport from the ER to the Golgi apparatus.

The model of the endodomain revealed six β-strands, which formed two β-sheets with the same topology as in dimeric Gp5/M; β1, β2 and β6 built one sheet, and β3, β4 and β5 built the other sheet. Six β-strands were also recently predicted for monomeric M and Gp5 using sequence similarity-based clustering and domain analysis [70]. This indicates that the seventh β-strand, which makes most of the contacts between Gp5 and M, forms upon dimerization from the C-terminal region (Figure 7D).

### 2.10. Gp5/M Has a Similar Structure as the Orf3a and M Proteins of SARS-CoV-2

Querying the protein structure database for homologs of Gp5/M with Dali (http://ekhidna2.biocenter.helsinki.fi/dali/, accessed on 6 June 2022) returned only weak hits; one hit, however, was the Orf3a protein of SARS-CoV-2 [71]. This might be of relevance since Corona- and Arteriviruses belong to the same virus order, the Nidovirales, and hence an evolutionary relationship exists. In addition, the recently determined structure of the M of SARS-CoV-2 revealed a similar folding as Orf3a [72]. The M protein is the most abundant structural protein of coronaviruses and is considered the major driver of virus assembly and membrane budding [72]. In contrast, Orf3a is a non-selective cation channel that is only a minor component of virus particles [73].

The structures of Orf3a and M, which are homodimers, exhibited (almost) the same folding as the Gp5/M heterodimer with an N-terminal 3-TM region and a C-terminal β-sandwich domain (Appendix A). A representation of the molecular surface revealed that the endodomain of Orf3a and M were more compact and closer to the TMs, so there was not a large cavity with openings on two sides, as was the case with Gp5/M. However, the region between the TMs and the endodomains was highly basic, which is another common feature of all three proteins (Appendix A). The six TM helices of all three dimers did not run straight through the membrane, but at an angle. Due to the presence of a proline (wheat spheres in Figure 8), both M and Gp5/M (but not Orf3a) contained a kinked helix, which is TM1 in Gp5/M and TM2 in the M of SARS-CoV-2. Two different conformations have been described for the M of SARS-CoV-2: a short form where the monomers are connected by hydrophilic interactions in the outer part of the molecule, and a short form that lacks these interactions. The conformational change is due to a shift in the interaction of an amino acid in a hinge region following TM3 with a residue in TM2 of the same monomer. E115 in the hinge connects to Y47 in the short form, but to K50 in the long form of M [72]. A similarity between Gp5/M and Orf3a are palmitoylated cysteines at the C-terminus of TM3 (magenta spheres in Figure 8, see Section 2.11. below). The M of SARS-CoV-2 contains no cysteines in this region and the C-terminus of TM3 is not amphiphilic.

All three dimers exhibited hydrophilic interactions between the monomers within the membrane, but their position was not conserved (green and cyan spheres in Figure 8). The difference in the location of the hydrophilic interactions probably caused the spatial orientation of the three transmembrane helices to be different for Gp5/M, M and Orf3a. The three TMs of Gp5 and M could be well superimposed on the six TMs of Orf3a, except that the TM1 and especially TM2 of Gp5 and M are shorter (Figure 9A). The alignment of Gp5/M with M of SARS-CoV-2 was less pronounced, in part due to the highly curved TM1 helix of Gp5/M (Figure 9B).

The endodomains of M and Orf3a formed a pair of opposing ß-sheets packed against one another in an eight-stranded ß-sandwich; one ß-sheet contained ß-strands ß1, ß2, ß6 and ß7, and the other sheet contained ß3, ß4, ß5 and ß8. The endodomain of Gp5/M contained only seven ß-strands, but they were arranged in a similar order. The endodomains of the M of PRRSV aligned substantially with the endodomains of the Orf3a and M of SARS-CoV-2, especially ß3, ß4, ß5 and a part of ß7, which formed one of the two ß-sheets. The C-terminal parts of ß7 then ran apart, and the ß7 of the Orf3a and M of SARS-CoV-2 was connected by a short loop to ß8, which was not present in the M or Gp5 of PRRSV (Figure 9C,D).

We then performed pairwise and multiple amino acid sequence alignments between the Orf3a and M of SARS-CoV-2 and the Gp5 and M of the Arteriviruses. The highest homologies, as expected, were calculated between M (78–91%) and Gp5 (61–72%) of both PRRSV viruses (Appendix A). However, significant homologies were also calculated if the Gp5 proteins were compared with the M proteins of each Arterivirus (10–33%, more pronounced in a pairwise alignment). Likewise, the M of the Arteriviruses exhibited a homology of 15–30% to the M of SARS-CoV-2 and 11–14% to Orf3a. The homology values were in the same range when the Gp5 of the Arteriviruses was compared with the M (13–29%) and Orf3a (4–19%) of SARS-CoV-2. Amino acid similarities of about 30% (or more) caused the respective proteins to fold in the same way. Values between 20% and 30% were in a grey area where the same folding was possible but not certain. This suggests that M, Gp5 and Orf3a may have descended from a common ancestor present in an ancient member of the order Nidovirales before it diversified into the Coronaviridae and Arteriviridae families.

Orf3a is a non-selective cation channel that contains a large polar cavity between the TM regions. The cavity is connected to the molecule`s surface by several tunnels, which can be visualized with the tool MOLE 2.5 (http://beta.mole.upol.cz/, accessed on 26 June 2022). The upper tunnels are formed between TM2 and TM3 within each protomer and likely open to the membrane. The lower tunnels run underneath the TM1–TM2 linker and above the endodomain, and open to the cytosol. By applying the same algorithm to the Gp5/M dimer, we identified just one small tunnel between the TM1 and TM2 of Gp5, which was lined with mostly hydrophobic residues (Appendix A). Based on this analysis, it seems unlikely that the Gp5/M dimer functions as an ion channel. Likewise, the structure of the M of SARS-CoV-2 also only has small hydrophobic tunnels, making it unlikely that water molecules or ions could approach them [72].

### 2.11. Orf3a Is Palmitoylated at a Cluster of Cysteine Residues near the C-Terminal Part of TM3

Another common feature of Gp5/M and Orf3a is a cluster of cysteine residues near the C-terminal part of TM3. Orf3a contained five cysteines in close proximity in the 3D structure: two were located at the end of TM3 (Cys130, Cys133) and three were in the adjacent cytoplasmic region (Cys148 in β1, and Cys153 and Cys157 are both located in the loop between β1 and β2 (Figure 10A)). Most of them were exposed at the surface of the molecule and hence could be subjected to fatty acid attachment. Except for Cys153, these residues were also present in the Orf3a of SARS-CoV. Conserved between SARS-CoV and SARS-CoV-2 was also Cys81 in the middle of TM2. To test whether one or more of these cysteines were acylated, they were individually replaced with serine, and the resulting constructs, fused to a C-terminal V5 tag, were expressed in 293T cells. Forty-eight hours after the transfection, cells were lysed and subjected to Acyl-RAC (resin-assisted capture), which exploits thiol-reactive resins to capture SH groups in proteins. A total of 10% of the total extract (TE) was removed from the lysate to determine the expression levels of Orf3a. Disulfide bonds in the proteins present in the remaining part were reduced, and newly exposed -SH groups were blocked. The sample was then equally split: one aliquot was treated with hydroxylamine to cleave thioester bonds, and the other aliquot was treated as a control with a Tris-HCl buffer. After the pull down of the proteins with the thiol-reactive resin, the samples were subjected to Western blotting using antibodies against the V5 tag. To exclude the proteins that were lost during sample preparation, we used antibodies against the cellular palmitoylated protein flotillin 2 as an internal control (Figure 10B). A strong signal was detected for Orf3a wt in hydroxylamine-treated (+HA) but not Tris-treated samples (-HA), indicating that the protein was acylated. The exchange of Cys81 had no effect on Orf3a expression (TE) and its palmitoylation level (+HA) compared with Orf3a wt. A replacement of the other cysteine residues reduced the palmitoylation of Orf3a but could not completely prevent it. Note, however, that the expression level of most Orf3a mutants is also lower, as has been previously observed for C133. The quantification of the bands revealed no reduction in the palmitoylation of the mutant C81S (96%), a moderate reduction in C133S (72%), a stronger reduction in C130S (51%) and C157S (53%), and the strongest reduction in mutants C148S (31%) and C153S (38%) relative to wt Orf3a (100%). We then created a mutant where four cysteines (130, 133, 148, 153) were exchanged simultaneously to analyze whether acylation was completely abolished. The signal for this mutant in the total cellular extract was much weaker, which prevented the determination of the palmitoylation level. The mutant Orf3a was apparently degraded since the amount was weaker at 48 h compared with 24 h after transfection, whereas the signals for Orf3a wt increased during this time period (Figure 10C). We conclude that Orf3a was acylated at several cysteines located at the end of TM3 and in the cytoplasmic tail, and that the cysteines worked synergistically with each other, i.e., acylation at one Cys increased the probability that a second residue was modified, as described previously for the Gp5/M of PRRSV and the S of SARS-CoV-2. Moreover, the removal of the acylation sites gradually destabilized Orf3a, causing its degradation.

## 3. Discussion

### 3.1. Role of the Gp5/M Ectodomain for Neutralization of PRRSV

The amino acids 5 to 12 that built the neutralizing antibody epitope in the Gp5 of the VR 2332 projected from the N-terminal helix, except for the side chain of residue 12, which was located on the flat surface of Gp5, close to residue 10 of M (Figure 3). An overlapping but larger epitope (residues 4–18) was identified in the Gp5 of the PRRSV-1 prototype strain Lelystad, but only the residues present in the N-terminal helix were located on the molecule’s surface (Figure 6). Contrary to what one may intuitively suspect, this epitope was well conserved between PRRSV-1 strains, and only a few amino acids of the epitope of PRRSV-2 were highly variable, which were, however, located on the underside of the N-terminal helix. The whole ectodomain of M in PRRSV-1 strains and residue 10 in the M of PRRSV-2 strains was almost more variable. The latter contained hydrophilic amino acids, mostly with large side chains, but different chemical properties, such as Asn, His, Tyr, Arg or (rarely) Met, and there is considerable evidence that these substitutions confer resistance against a broadly neutralizing antibody [37] (Appendix A, Appendix A). The most variable residues in the PRRSV-1 and PRRSV-2 strains were not part of a described antibody epitope. One hypervariable region was located at the N-terminus of Gp5, which is discussed below. Another hypervariable region was located at the C-terminal portion of the ectodomain of Gp5. The region contained a large variety of mostly large, hydrophilic residues that were exposed at the surface of the molecule opposite to the N-terminal helix. No function was attributed to this region, and the evolutionary forces that drive the frequent exchange of amino acids is also not known.

An immunodominant, but non-neutralizing decoy epitope was identified in the N-terminal portion of the Gp5 of PRRSV-2, which may not be present in the mature protein, depending on the signal peptide cleavage site [35]. Predicting the structure of the Gp5/M of VR 2332 cleaved at site Ala26/Val27 revealed that the additional five amino acids in Gp5 extended the N-terminal helix; otherwise, the structure remained unchanged (Appendix A).

Does this structure of the Gp5/M of VR 2332 support the concept that an antibody bound to the decoy epitope prevents the binding of another antibody directed against the neutralizing epitope? The distance between the first and the last amino acid of the N-terminal helix carrying both the “decoy” and the neutralizing epitope was 22 Å, while the approximate length of the antigen-binding surface of an antibody was larger, about 28 + −8Å [74]. Based on these rather crude numbers, it seems possible that both antibodies could compete for their binding sites, but whether this results in an effective displacement of the neutralizing antibody also depends on the concentrations of the two antibodies and their affinity for their epitopes. Note also that Gp5/M molecules having a decoy epitope are probably a minor fraction in VR 2332 virus particles, and hence the majority would be accessible to neutralizing antibodies [36].

### 3.2. Heterogeneity in Signal Peptide Cleavage Sites of Gp5 of PRRSV

To determine whether the Gp5 of other PRRSV-2 strains might retain the decoy epitope, we predicted the signal peptide cleavage sites of 180 Gp5 proteins of PRRSV-1 and 136 of PRRSV-2 strains. The results revealed that although the majority of them (75%, 87%) were cleaved at the same site, between residues 32 and 33 and 31 and 32, respectively, the mean probability for all the predictions was low, at 54% and 47%. Furthermore, especially for the Gp5 proteins with a low probability for cleavage at the main site, a second or third minor cleavage site was predicted (Appendix A). This is consistent with a mass spectrometry analysis of purified Gp5 of VR 2332, which is cleaved mainly at site Ala31/Ser32 but also at site Ala26/Val27 [36]. N-terminal peptide sequencing revealed Ser32 as the first amino acid in the Gp5 of VR 2332 virions, but this method is less sensitive than mass spectrometry, and hence the minor Gp5 fraction retaining the decoy epitope might have been missed [75]. Furthermore, for some Gp5 proteins, the probability that the N-terminus functioned as a signal peptide was rather low. This might explain why signal peptide removal from some Gp5 proteins was not very efficient if expressed from a plasmid but could be rescued by the co-expression of M [67].

Whether the variation in the cleavage of the signal peptide has any functional significance is not known. However, some amino acids of the signal peptide are apparently under evolutionary pressure. A large number of sites, residues 3, 11, 13, 14, 20, 21 and 29, were reported to be positively selected after PRRSV emerged in swine after cross-species transmission from an unknown host [76]. Likewise, a rebound PRRSV-2 virus exhibited one positively selected site (Ala29Val) in the signal peptide, increasing the likelihood of cleaving at a second site [77]. It is difficult to understand why amino acid substitutions in the signal peptide are positively selected in pigs when its only function is to direct Gp5 synthesis to the ER membrane.

The variation in signal peptide cleavage not only determines whether a decoy epitope is present in the mature Gp5 protein, but also whether the N-terminal region contains additional N-glycosylation sites (Figure 3 and Figure 6). We showed that such an additional N-glycosylation site is at least partially used in the subfraction of the Gp5 molecules of VR 2332 with a longer N-terminal helix [36]. Likewise, the short but highly variable region following the main cleavage site in Gp5 of both the PRRSV-1 and -2 strains contains mainly N, S or T residues and thus determines whether another carbohydrate is added. Gp5 molecules from some strains even contain the sequence NNSS. For some proteins, including the Gp3 of EAV, it was demonstrated that both positions are (at least partially) filled with a carbohydrate [78]. The additional carbohydrates are located on top of the N-terminal helix of Gp5 and thus are in an ideal position to shield the neutralizing epitope in the helix. Single amino acid exchanges occurring during virus evolution are then sufficient to add or remove a carbohydrate that could hide or expose the antibody epitope. Thus, glycan shielding might explain antibody escape in the absence of a large variation in amino acids in the antibody epitope, as previously suggested [44,45,46].

### 3.3. The Function of the Transmembrane Region in Virus Assembly and Budding

Gp5/M is the driving force for virus budding, which involves the conversion of a planar membrane into a spherical vesicle. To do so, Gp5/M has to exert a pushing force on the lipid bilayer to form an Ω-shaped bud, which then needs to be cut off from the membrane to release a virus particle [65]. The overall shape and the particular structure of the transmembrane regions of Gp5/M with their kinked and twisted helices could play a decisive role in this process (Figure 4 and Figure 6). For energetic reasons, the main components of the hydrophobic part of a lipid bilayer, the acyl chains of phospholipids and cholesterol, like to arrange themselves parallel to each other and also to the transmembrane region. This arrangement is locally disrupted if the transmembrane regions do not run perpendicularly through the membrane, making the bilayer more susceptible to curvature formation [79].

Moreover, a Gp5/M dimer has the shape of an inverse cone; the external part of the transmembrane region is smaller than the internal, cytoplasmic part, with the two amphiphilic extensions of the TM3 of both Gp5 and M (Figure 2C and Figure 8C). A protein with such a shape could cause negative curvature of the membrane, i.e., the lipid bilayer curves inward toward the cytoplasm. In principle, virus budding requires mainly positive membrane curvature, but the neck region of the Ω-shaped bud is a region of negative curvature [80,81].

The amphiphilic helices at the C-termini of the helix 3 of the Gp5 and M of PRRSV-1 and -2 might be also instrumental in the release of virus particles (Figure 4 and Appendix A). They contain fatty acids in both Gp5 and M, and the acylated cysteines are present in the hydrophobic part of the helices, indicating that the acyl chains insert into the lipid bilayer. A helix with similar biophysical properties is present in the M2 protein of the Influenza A virus, which inserts like a wedge into the inner leaflet of the bilayer [82]. This leads to the splaying of lipids in one leaflet, which curves the membrane and catalyzes the release of virus particles. One could imagine a similar role for the amphiphilic helices present in both the Gp5 and M of the PRRSV-1 and PRRSV-2 strains. However, the single acyl chain attached to the amphiphilic helix of M2 plays a minor role in virus replication [83]. This is in contrast to the three and two fatty acids attached to Gp5 and M, respectively, that are essential for virus budding and replication [30]. Mechanistically, protein-bound fatty acids might drive the formation of cholesterol and sphingolipid-rich lipid nanodomains at the viral budding site, as has been demonstrated for the spike of SARS-CoV-2 [84]. This could promote the formation of larger domains, which eventually detach from the membrane due to “line tension”, similar to what has been proposed for the budding of viruses, from raft domains at the plasma membrane [85]. In summary, Gp5/M contains several elements that could act on a lipid bilayer, but how exactly and in what temporal sequence they work remains to be determined.

Most residues in the transmembrane helices were highly conserved between virus strains (Appendix A). This is rather unusual for transmembrane regions, which in principle could not only accommodate the eight hydrophobic amino acids, but also others, such as Ser, Thr, Cys, Gly and Tyr (95). This sequence conservation speaks in favor of a specific function of individual amino acids beyond mere membrane anchoring. The TM2 of Gp5 interacts with glyceraldehyde-3-phosphate dehydrogenase (GADPH), an enzyme involved in glycolysis. Binding prevents GAPDH’s translocation from the cytosol to the nucleus and facilitates PRRSV replication by virtue of its glycolytic activity [86]. The binding site forms a bulge on the outside of Gp5 that encompasses the outer (lumenal) part of TM2, including the following surface-exposed loop (Appendix A). Interestingly, two of the amino acids, residues 64 and 75, which are involved in the binding of GAPDH, were positively selected upon the spill over of PRRSV from an unknown host to pigs, suggesting that it represents an adaption to porcine GAPDH [87].

### 3.4. A putative Role of the Endodomain in Genome Recruitment and as a Protein Binding Site

A long-standing model for virus assembly proposes that a membrane protein must bind to the nucleocapsid in order to recruit it to the viral budding site [88]. The nucleocapsid of Arteriviruses consists of a double-layered chain of N proteins that contain the viral genome in their center. The positively charged N-terminal domain of N binds the viral RNA, and the C-terminal domain is supposed to interact with viral envelope proteins [47]. Its crystal structure reveals a dimer that is composed of a four-stranded β-sheet floor that is superposed by two α-helices and thus resembles the peptide-binding site of an MHC molecule [89]. A surface projection of the N dimer viewed from above shows a groove with two circular surface indentations between the two α-helices. Each helix exposes four hydrophilic amino acids that could interact with other proteins (Appendix A). It was proposed that this structure is at the outside of the nucleocapsid and is the binding site for the cytoplasmic tails of Gp5 and/or M [7,47].

However, the structure of the Gp5/M dimer does not directly reveal which part of its endodomain might bind to N. The C-termini of both proteins interact with each other, limiting their capacity to interact with other proteins. In principle, amino acids in the loop between β3 and β4, which contain basic residues in both M and Gp5, are surface exposed and hence could bind to N (Appendix A). We used the HDOCK server (http://hdock.phys.hust.edu.cn/, accessed on 2 May 2022) to predict the putative protein–protein interactions between N and the structure of β3 and β4 including the loop. The algorithm suggested several models for Gp5 and M, where the side chain of a basic amino acid is submerged into the larger indentation. One for M is shown in Appendix A, where Arg137 forms hydrogen bonds with the main chain of both α-helices. The other hydrophilic residues in the loop form various interactions with the exposed amino acids at the α-helices. The rather low number of electrostatic interactions suggests that the affinity of a single interaction between Gp5/M and N is weak. However, avidity, the accumulated strength of multiple interactions between a large number of Gp5/M dimers in the membrane and N dimers in the cytosolic nucleocapsid, increases the total strength of the interaction. Cryo-EM tomographs of PRRSV particles revealed that the virus envelope was separated from the nucleocapsid by a 2–3nm gap, which was traversed only by a few strands of density [47]. This indicates that the interactions between envelope proteins and the capsid are weak and flexible and are disrupted after virus budding. Note, however, that this model is highly speculative and needs to be confirmed with experimental data.

The most striking feature of the Gp5/M dimer is the positively charged, perforated cavity between the membrane and the endodomain (Figure 5 and Figure 6). Its function is unclear, but one might assume that it serves as an interaction site with other proteins having an acidic surface. Since the sequences of the other viral membrane proteins, Gp2, E and Orf5a, have mostly basic residues in their endodomain, they are unlikely to interact with the basic endodomain of Gp5/M. Gp3 also does not qualify as an interaction partner since it is peripherally anchored in a hairpin-like structure in the outer lipid layer of the membrane [78,90]. Likewise, the Gp4 of PRRSV-1 and -2 have only very few amino acids exposed to the cytosol. Recently, it was reported that nsp2TF, a C-terminally modified variant of the non-structural protein 2a, binds to the Gp5/M dimer. The deubiquitinating activity of nsp2TF removes ubiquitin from the Gp5/M proteins, thus preventing their proteasomal degradation [91]. Nsp2TF is anchored by four hydrophobic regions at its C-terminus to membranes of the exocytic pathway. The N-terminal part, which is exposed to the cytosol, is acidic (it has a calculated isoelectric point of 5.4) and hence might bind to the basic region in Gp5/M. Another possible candidate for a cellular protein interacting with the Gp5/M endodomain is monomeric actin, which has an acidic surface. Actin is involved in the budding of many enveloped viruses and actin polymerization is known to induce membrane curvature, but its exact role has not been elucidated [92,93]. Moreover, 122 and 219 cellular proteins were identified by mass spectrometry in samples immunoprecipitated from PRRSV-infected cells with Gp5- and M-specific antibodies, respectively [94,95].

### 3.5. Gp5 and M Are Members of a Protein Superfamily That Includes M and Orf3a-like Proteins from Coronaviruses

The Gp5/M dimer has a very similar fold as the Orf3a and M of β-coronaviruses, and their transmembrane domains and endodomains superimpose quite well (Figure 8, Figure 9 and Appendix A). The similarity is corroborated by the relatively high amino acid similarity between all three proteins (Appendix A). Recently, based on sensitive homology searches, three other Orf3a-like proteins were uncovered and prototyped by the ORF5 of MERS-CoVs, ORF4 of 229E-related bat CoV and ORF3 of Eidolon bat coronavirus [70,96].

We also show here that the Orf3a of SARS-CoV-2 is S-acylated at a group of cysteine residues at the end of TM3 and in the adjacent cytoplasmic tail, which is another common feature with Gp5/M (Figure 10). Almost any Orf3a-like protein contains at least one, but mostly several cysteines in this region, but not always at the same position [70]. It is not uncommon for viral membrane proteins that acylated cysteine residues shift their position during virus evolution [97]. One of the cysteines of Orf3a, which we have shown here to be acylated (Cys133), was previously proposed to form disulfide bonds with each other, leading to the dimerization and tetramerization of Orf3a [98]. However, the elucidation of the structure of an Orf3a dimer revealed that it was not connected by disulfide bonds, and Cys133 was located on the outside of the dimer [71]. Disulfide bond formation between cytoplasmic cysteines is very rare anyway because the cytosol is a reducing environment [99]. Nevertheless, the area around the cysteine-rich domain might be involved in tetramerization by non-covalent hydrophobic interactions, and the covalently bound fatty acids might contribute to that. It was shown that the replacement of Cys133 reduces the expression of Orf3a [98]. We show here that the acylation of Cys133 (and of other cysteines) destabilizes Orf3a, as the mutants are degraded more rapidly in transfected cells (Figure 9).

Orf3a is a non-selective calcium-permeable cation channel that conducts ions via a central polar cavity that opens to the cytosol and membrane through separate water- and lipid-filled openings [71,98]. Similar hydrophilic tunnels are apparently not present inside the Gp5/M structure of PRRSV (Appendix A) and M of SARS-CoV-2 [72], and hence it seems unlikely that these proteins function as ion channels. In addition, in contrast to Gp5/M, Orf3a is not a (or at least not a major) component of virus particles and it is not essential for virus replication in cell cultures, and hence it is not the main driver for virus budding [73].

This function is attributed to the M protein of coronaviruses and the Gp5/M of Arteriviruses. Both are essential for virus replication, are the major component of virus particles and the driving force of virus assembly and budding at internal membranes on the exocytic pathway. The M of SARS-CoV-2 binds to the viral nucleocapsid protein [72], and the same role was discussed for Gp5/M (Section 3.4). Both Gp5/M and M dimers have the shape of an inverse cone, which might be instrumental to induce membrane curvature. They also contain one kinked helix, TM1 in Gp5/M and TM2 in the M of SARS-CoV-2 (Figure 8 and Figure 9), which might disturb the parallel orientation of the membrane lipids. However, the spatial orientation of the three transmembrane helices is different in the Gp5/M and M of SARS-CoV-2, partly because the hydrophilic interaction between the monomers is at a different location (Figure 8). Essential for PRRSV budding is fatty acid attachment to conserved cysteine residues at an amphiphilic helix at the C-terminus of the TM3 of Gp5 and M [30]. The M of SARS-CoV-2 does not contain cysteines in this region, which is also not amphiphilic. However, the M proteins of various bat viruses, infectious bronchitis virus and MERS do [70], and hence M might also be acylated in some coronaviruses. In summary, the exact way that virus budding is induced might differ between Gp5/M and M.

Since Gp5/M and Orf3a apparently perform different functions, it seems unlikely that their similar structures have arisen by convergent evolution. More likely, all Gp5-, M- and Orf3a-like proteins originated from the same gene that was present in a common ancestor of the Nidovirales superfamily and was involved in budding, a very basic feature of virus replication. Before the ancestor split into the Corona- and Arteriviruses, the gene was duplicated, which generated the genes encoding M and Gp5 in Arteriviruses and M and Orf3a in Coronaviruses. In agreement with this, it has been suggested recently that ORF3a may be derived from the M gene by gene duplication [70,96]. Likewise, the duplication of Arterivirus genes is not unprecedented, since the Simian hemorrhagic fever virus, which is the Delta Arteriviruses reference strain, contains two copies of the genes encoding the minor glycoproteins Gp2, Gp3 and Gp4 [100].

In summary, we obtained a reliable and accurate structure of a complete Gp5/M dimer of PRRSV-1 and -2. To our knowledge, except for some SARS-CoV-2 proteins (https://www.deepmind.com/open-source/computational-predictions-of-protein-structures-associated-with-covid-19, accessed on 24 June 2022), this is the first time that a in most parts reliable structure of an oligomeric viral protein has been predicted de novo with alphafold2. However, alphafold2 cannot yet successfully predict every viral protein, as we have not been able to obtain a reliable structure for the membrane proteins Gp2, Gp3 and Gp4 of PRRSV. The confidence values are crucial to assess whether the predicted models (and which parts of it) are reliable. The Gp5/M model allows one to not only put the sparse data on the function of Gp5/M on a structural basis, but also to make experimentally testable hypotheses about its role in virus replication and as a target for an antibody-mediated immune response. The results also showed that alphafold2 can be successfully used to predict the structures of viral membrane proteins from their amino acid sequence alone, even if they are oligomers. This is likely facilitated by a large number of different sequences in the databases, as it enables evolutionary covariance analysis, which is the basis of alphafold2.

## 4. Materials and Methods

### 4.1. Predictions of Protein Structures by Alphafold2

For the prediction of the dimeric version of the Gp5/M of PRRSV-1 Lelystad, PRRSV-2 VR 2332, we used the alphafold2advanced.ipynb notebook [101] (https://colab.research.google.com/github/sokrypton/ColabFold/blob/main/beta/AlphaFold2_advanced.ipynb, accessed on 24 and 25 October 2021) with the following settings: msa_method = mmseqs2, homooligomer = 1:1, pair_mode = unpaired, max_msa = 512:1024, subsample_msa = True, num_relax = 1, use_turbo = True, use_ptm = True, rank_by = pLDDT, num_models = 5, num_samples = 1, num_ensemble = 1, max_recycles = 3, tol = 0, is_training = False and use_templates = False.

For the prediction of the monomeric version of Gp5 and of PRRSV-1 Lelystad and PRRSV-2 VR 2332, we used the ColabFold: Alphafold2 using MMseqs2 notebook https://colab.research.google.com/github/sokrypton/ColabFold/blob/main/AlphaFold2.ipynb, accessed on 9 October 2021 with the following settings: num_models = 5, use_amber = False, use_msa = True, msa_mode = MMseqs2 (UniRef + Environmental), use_templates = False and homooligomer = 1.

All the notebooks used MMseqs2 instead of Jackhammer in the DeepMind original notebook, which is a fast and sensitive method to search protein databases to build the multiple sequence alignment [102]. All Colab notebooks are accessible for free for users logged in with a Google account, and they include access to powerful Graphical Processing units (GPU) and/or Tensor Processing Units (TPU). The predictions are freely available for both academic and commercial use under Creative Commons Attribution 4.0 (CC-BY 4.0) license terms.

For structure predictions we used amino acid sequences from the prototype PRRSV-1 strain Lelystad (GenBank: M96262.2) and the prototype PRRSV-2 strain VR 2332 (AY150564.1), lineage 2.1. For structure predictions of the representative members of four other PRRSV-2 lineages [61], we used the amino acid sequences of isolate HK2, lineage 2.2, KF287133.1; isolate MN414, lineage 2.6, KT581982.1; strain CH-1a, lineage 3.1, AY032626.1; and strain WUH3, lineage 3.7, HM853673.2 in the case of Gp5 without their signal peptides. Since signal peptides are usually rapidly cleaved after the translocation of the protein into the lumen of the ER, they are unlikely to affect protein folding. In the sequences of the Gp5 of VR 2332 and Lelystad, we removed the first 31 and 34 residues, respectively, which were the main cleavage sites determined by mass spectrometry [36].

### 4.2. Assessment of the Quality of the Model of the Gp5/M Dimer

Alphafold2 generates a PDB file, which can be analyzed with standard molecular visualization systems, such as PyMol, and two confidence scores. The first confidence score is a per residue confidence metric called the predicted local distance difference test (pLDDT), which indicates the confidence in the local structure prediction. The scale ranges from 0 to 100 and an IDDT value above 90 indicates very high accuracy, equivalent to structures determined by experiments, which allows for the investigation of details of individual side chains. A value from 70 to 90 indicates high accuracy, where the predictions of the protein’s backbone are reliable. A value of 50 to 70 indicates lower accuracy, but the predictions of the individual secondary structural elements, α-helices and β-strands are probably correct, but how they are aligned in space is uncertain. Values below 50 might be an indication of an intrinsically unstructured region. The PDB files with the predicted structure contain this information in the B-factors, which can be highlighted in the 3D protein structure. Areas with high B-factors, which indicates high confidence, are colored red, while low B-factors are colored blue. The second confidence metric, called the prediction aligned error (PAE), measures confidence in the relative positions of pairs of residues. The PAE is displayed as a 2D plot, and the expected position error in Angström is color coded. The expected position error is usually low for two amino acids present in one domain, but it is sometimes high for residues located in two domains. This indicates that Alphafold is uncertain about the relative position of two domains that have no predicted contact with each other, for example, if they are connected by a flexible linker. Alphafold2 also provides a graph showing how many sequences were used for the initial multiple sequence alignment (MSA) and which parts of the full-length protein were covered by the analysis.

The assignment of the secondary structural elements to the amino acid sequence was conducted with POLYVIEW-2D (http://polyview.cchmc.org/, accessed on 10 February 2022) by submitting the PDB file for each predicted structure to the server.

To analyze various other parameters of the predicted Gp5/M structure, such as the Ramachandran plot, ionic bonds and inter-subunit contacts, we used PDB-sum, http://www.ebi.ac.uk/thornton-srv/databases/pdbsum/Generate.html, accessed on 10 February 2022 [103].

### 4.3. Visualization and Analysis of Predicted Gp5/M Structures

We mainly used PyMol version 2.1.1 (Molecular Graphics System; Schrödinger, LLC: New York, NY, USA) to visualize and analyze the PDB files made available by alphafold2. Some figures were created with ChimeraX 1.3 (https://www.rbvi.ucsf.edu/chimerax/, accessed on 6 November 2021) as indicated. The APBS (Adaptive Poisson–Boltzmann Solver) electrostatics plugin for Pymol was used to calculate and display the electric charges on the surface of the Gp5/M dimer [104]. Note that the numbering of the amino acids in the figures starts with the first amino acid of the mature protein, i.e., without its signal peptide. To calculate the respective position in the complete Gp5, the length of the signal peptide must be added, which is 31 for the Gp5 of VR 2332 and 34 for the Gp5 of Lelystad.

### 4.4. Calculation of the Position of Gp5/M within a Virtual Lipid Bilayer

To orient the Gp5/M structure in a virtual lipid bilayer, we used the PPM 3.0 Web Server https://opm.phar.umich.edu/ppm_server3, accessed on 2 April 2022. The location of a protein in the membrane coordinate system is obtained by the optimization of protein transfer energy (ΔG transfer) from water to a lipid bilayer. In other words, as many hydrophobic amino acids as possible should be inside the hydrophobic part of the bilayer to avoid a hydrophobic mismatch. The anisotropic properties of the lipid bilayer composed of 1,2-dioleoyl-sn-glycero-3-phosphocholine (DOPC) are described by transbilayer profiles of dielectric constant and hydrogen bonding acidity and basicity parameters [105]. The type of membrane was set to a flat Golgi membrane since this is the main localization of the Gp5/M inside cells. The Golgi membrane had a hydrophobic thickness of 30.2 ± 1.3 Å, which roughly corresponds to the length of two acyl chains. Note that the whole bilayer was thicker (~10 Å) since the hydrophilic head groups of the lipids were outside the hydrophobic core domain [76]. As input for the program, the PDB files of model 1 of the Gp5/M dimer of PRRSV VR 2332 and Lelystad were used. The program provided a PDB file of the dimer positioned in the membrane, with the boundaries of the hydrophobic domain marked by dummy atoms, shown here as grey spheres. A ΔG transfer of −61.6 kcal/mol was calculated for both structures, which lied within the range between −400 and −10 kcal/mol usually calculated for integral membrane proteins.

### 4.5. Sequence Conservation Analysis

The 1077 ORF5 nucleotide sequences encoding Gp5 from the PRRSV-1 strains and 7701 from the PRRSV-2 strains, and 83 ORF6 nucleotide sequences encoding M from the PRRSV-1 strains and 290 from the PRRSV-2 strains present in the NCBI database were translated into the corresponding amino acid sequences and were aligned using Clustal W. From these data, a consensus sequence for the Gp5 and M of both PRRSV-1 and PRRSV-2 containing the most abundant amino acid at each position was compiled, and the percent conservation at each position was plotted against the amino acid number.

To determine and visualize which amino acids are present at a certain position of the sequence and at which frequency, we used web logo http://weblogo.threeplusone.com/, accessed on 24 May 2022. The web-based application generates a graphical representation from a multiple sequence alignment. Each logo consists of stacks of amino acid symbols, one stack for each position in the sequence. The overall height of the stack indicates the sequence conservation at that position, while the height of the symbols within the stack indicates the relative frequency of each amino at that position [106]. To keep the numbering of the positions in the web logo identical to the consensus sequences, we removed the few sequences with insertions from the multiple sequence alignment before it was submitted to the web logo analysis.

To compare the Gp5 and M sequences of PRRSV with each other and also with the M and Orf3a of SARS-CoV-2, we performed both a pairwise sequence alignment using emboss needle https://www.ebi.ac.uk/Tools/psa/emboss_needle/ and a multiple sequence alignment using Clustal Q https://www.ebi.ac.uk/Tools/msa/clustalo/, both accessed on 22 Jume 2022.

### 4.6. Prediction of Signal Peptide Cleavage Sites in Gp5 of PRRSV-1 and PRRSV-2

We used SignalP 5.0, a deep-neural-network-based approach, to predict signal peptide cleavage sites from Gp5 [107]. Supplied with the full-length protein sequence, SignalP provided the information on whether or not the N-terminus of Gp5 acted as a signal peptide and also the position at which it was cleaved. The resulting summary sheets listed the Uniprot ID for each queried Gp5 protein, the prediction of whether the N-terminus is a eukaryotic signal peptide, the prediction of the cleavage site, the five amino acids surrounding the site and the probabilities for each prediction between 0 and 1. The mean and standard deviation of the probabilities and the highest and lowest probabilities for each cleavage site were also calculated. For each predicted sequence, a graphical representation was also delivered, which gave the probability that each of the ~70 N-terminal amino acids was or was not part of a eukaryotic signal peptide, and also whether it functioned as a cleavage site. In this way, additional putative cleavage sites can be identified with a lower probability than the main cleavage site.

Since SignalP5.0 cannot process thousands of sequences, we performed a protein blast search with the Gp5 sequence from the PRRSV-1 reference strain Lelystad and the PRRSV-2 reference strain VR 2333, respectively, against the non-redundant GenBank database CDS translations + PDB + SwissProt + PIR + PRF, excluding environmental samples with quick blast, which retrieved 136 Gp5 sequences of the PRRSV-1 strains and 180 Gp5 sequences of the PRRSV-2 strains.

### 4.7. Predicting the Amphiphilic Properties of the C-Terminus of Transmembrane Helix 3

We used the heliquest tool (https://heliquest.ipmc.cnrs.fr/, accessed on 8 March 2022) to calculate the physicochemical properties of the C-terminal part of TM3, which extends into the cytoplasm. The tool calculates two parameters: the hydrophobic moment (<µH>) and the average hydrophobicity (<H>) of a helix. The hydrophobic moment quantifies amphipathicity as the mean vector sum of the hydrophobicities of the side chains if this region forms an α-helix, whereas the hydrophobicity describes the avidity of the helix for lipids [108]. The program creates a helical wheel with an arrow inside, pointing to the hydrophobic side of the helix, and its length corresponds to the hydrophobic moment [109].

### 4.8. Docking of a Peptide from the Endodomain of Gp5/M to the Viral N-Protein

We used the HDOCK server (http://hdock.phys.hust.edu.cn/, accessed on 2. May 2022) to analyze possible interactions between the β-sheet 3 and 4, including the connecting loop in the endodomain of Gp5 and M with the N-protein of PRRSV [110]. The PDB file 1P65 containing the crystal structure of N was used as an input receptor molecule [89]. A PDB file was created from the Gp5/M VR 2332 prediction (model 2) containing only β-sheet 3 and 4, and it included the loop from either Gp5 or M by deleting all the other structural information from the PDB file, which was then used as the input ligand molecule. The server created several models where a basic amino acid of either Gp5 or M was inserted into the indentation in the surface groove of N. One of the models for the M-N interaction is shown in Appendix A. The other models showed other basic residues inserted into the indentation and/or showed other amino acids in the loop of Gp5 or M that interact with the amino acids in the α-helices of the N surrounding the surface groove.

### 4.9. Analysis of Orf3a and Gp5/M for the Presence of Cavities, Tunnels and Pores

We used Mole 2.5 (http://beta.mole.upol.cz/, accessed on 26 June 2022), a universal toolkit for the fully automated location and characterization of channels, tunnels and pores in (bio)macromolecular structures. After submitting the original PDB file, the program provides a PDB file with the location of tunnels, which can be overlaid with the structure of the respective protein plus other information about the dimensions of the tunnels and the limning amino acids.

### 4.10. Mutagenesis, Expression and Acylation Analysis of the Orf3a Protein of SARS-CoV-2

The pGBW-m4134028 mammalian expression plasmid containing the SARS-CoV-2 ORF3a protein with V5 Tag at the C-terminus was purchased from Addgene (#152048). Cysteines at positions 81, 130, 133, 147, 153 and 157 were exchanged to serine using the Strategene quick-change site directed mutagenesis protocol. All the mutations were confirmed by sequencing. Expression plasmids were used for the transfection of 293T cells (ATCC, CRL-3216) using the Lipofectamine 3000 according to the manufacturer’s instructions. The acylation of Orf3A was analyzed with an Acyl-RAC assay. The transfected cells were washed with PBS, lysed in 500 µL buffer A (0.5% TritonX100, 25 mM HEPES (pH 7.4), 25 mM NaCl, 1 mM EDTA, and a protease inhibitor cocktail). The disulfide bonds were reduced by adding Tris (2-carboxyethyl) phosphine (TCEP, Carl Roth) to a final concentration of 10mM, and they were incubated at RT for 30min. The free SH groups were blocked by adding methyl methanethiosulfonate (MMTS, Sigma, dissolved in 100 mM HEPES, 1 mM EDTA, 87.5 mM SDS) to a final concentration of 1.5% (*v*/*v*), and they were incubated for 4 h at 40 °C. Subsequently, 3 volumes of ice-cold 100% acetone were added to the cell lysate, and they were incubated at −20 °C overnight. Precipitated proteins were pelleted at 5000× *g* for 10 min at 4 °C. The pelleted proteins were washed five times with 70% (*v*/*v*) acetone, air dried, and then re-suspended in 1ml of a binding buffer (100 mM HEPES, 1 mM EDTA, 35 mM SDS). A total of 20–30 µL of the sample was removed to check for the total protein expression by Western blotting. The remaining lysate was divided into two equal volumes. One was treated with hydroxylamine (0.5 M final concentration, added from a 2M hydroxylamine stock adjusted to pH 7.4) to cleave thioester bonds. The other part was treated with 0.5M Tris-HCl pH 7.4. 30 µL thiopropyl agarose beads (Creative Biomart), which were beforehand washed by Millipore water, and it was added at the same time to capture the free SH groups. The samples were incubated with beads overnight at room temperature on a rotating wheel. The beads were then washed five times in a binding buffer and the bound proteins were eluted from the beads with a 2× reducing SDS-PAGE sample buffer for 5 min at 95°C. The samples were then subjected to SDS-PAGE and were immune-blotted using a rabbit polyclonal antibody against the V5 peptide (Genetex, # GTX117997, 1:3000) and mouse monoclonal anti-Flotillin-2 antibody (BD Biosciences, #610383, 1:1000). The secondary antibodies were coupled to HRP (anti-rabbit, Abcam, ab191866, 1:5000) and anti-mouse (Biorad, 1706516, 1:2000). All the antibodies were diluted in 5% non-fat dry milk. After washing, signals were detected by chemiluminescence using the ECL plus reagent (Pierce/Thermo, Bonn, Germany) and a Fusion SL camera system (Peqlab, Erlangen, Germany).

## Figures and Tables

**Figure 1 ijms-23-13209-f001:**
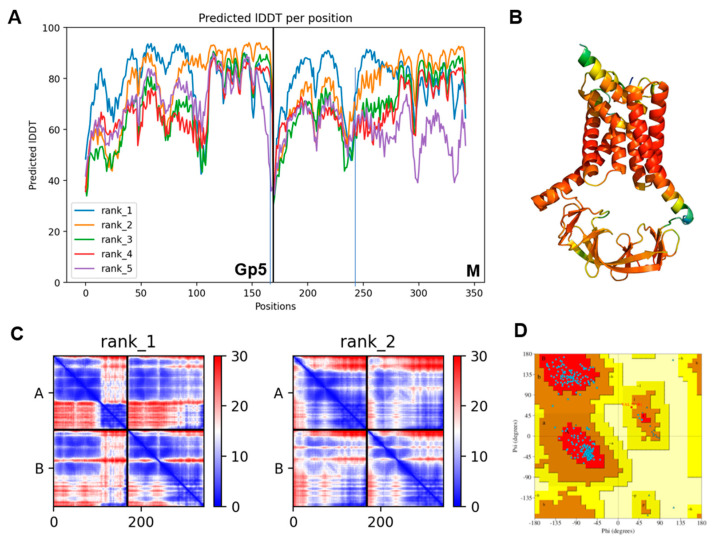
Confidence metrices for the predicted structure of Gp5/M of PRSSV-2 strain VR 2332: (**A**) Predicted local distance difference test (LDDT) score per position for the five models generated by alphafold2. The amino acid position is plotted against the predicted LDDT. Values between 70 and 90 indicate a high accuracy, where the prediction of the main chain of the protein is reliable. pLDDT values above 90 indicate very high accuracy, equivalent to structures determined by experiments. Values between 50 to 70 indicate a lower accuracy, but it is likely that the predictions of individual secondary structures are correct. (**B**) Cartoon model of the structure of Gp5/M showing the pLDDT per position, which is integrated as the b-factor in the PDB file delivered by alphafold2 in rainbow colors ranging from red (high confidence) to blue (low confidence). (**C**) Prediction aligned error (PAE) score for models ranked 1 and 2. This score displays the calculated error of the predicted distance for each pair of residues. Both axes indicate the position of the individual amino acids. The uncertainty in the predicted distance of two amino acids is color coded from blue (0 Å) to red (30 Å), as shown in the right bar. The color of the intersection of a horizontal line drawn from the position of an amino acid on the y-axis and a vertical line from the position of another amino acid on the x-axis indicates the error in the predicted distance between these two residues. PAE graphs are always characterized by a diagonal blue line, since amino acids that are juxtaposed in the primary sequence are also adjacent in the 3D structure. The upper left quadrant corresponds to errors in the distances of residues within Gp5, the lower right quadrant to errors within M, the upper right quadrant to errors between Gp5 and M, and the lower left quadrant to errors between M and Gp5. Note that model 1 has a lower error for the distances of the ectodomains, both within Gp5 and M and also between them. The errors for the transmembrane regions are very similar in both models. The error in the distances within and between the endodomains are also similar, but model 1 has a larger error in the distances between the endodomain and the transmembrane region. Note, however, that the differences between both models are rather marginal since both can be very well superimposed (Appendix A). (**D**) Ramachandran plot of the predicted structure of model 1. It shows the energetically allowed regions for backbone dihedral angles ψ against φ of amino acid residues in the Gp5/M structure. The φ-angle is the angle of the bond between the Cα-atom (which carries the amino acid side chain) to the N-atom, and the ψ-angle is the angle between the Cα-atom and the other C-atom of the peptide. For steric reasons, only certain combinations of ψ and φ angles are allowed. The red, brown, dark yellow and light yellow regions represent the favored, allowed, “generously allowed” and unallowed regions, respectively. Only one amino acid lies in the generously allowed region and none in the unallowed regions.

**Figure 2 ijms-23-13209-f002:**
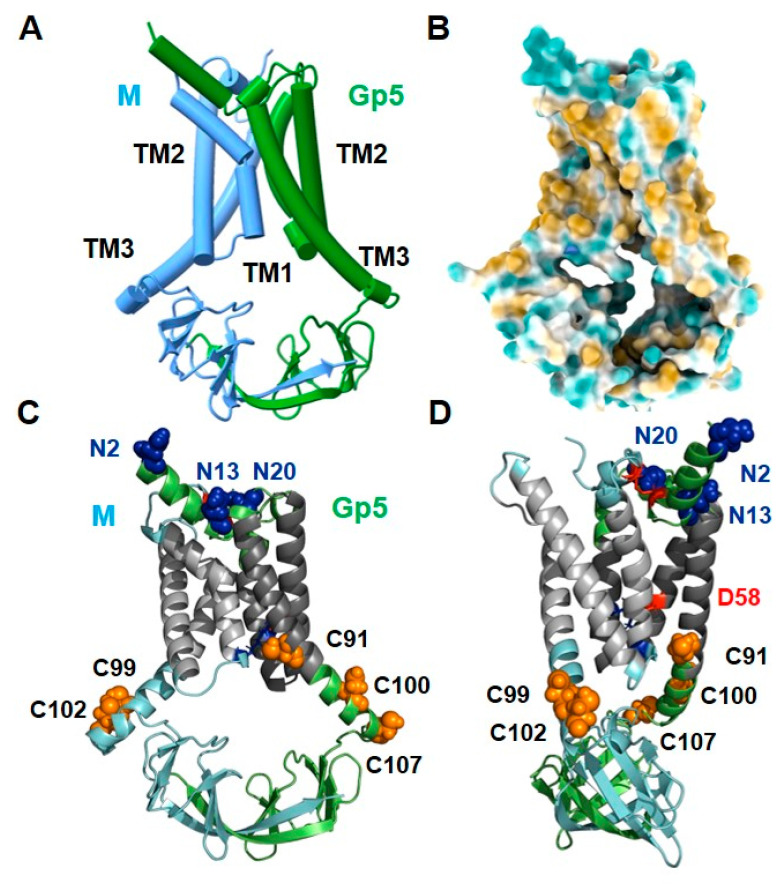
The predicted structure of Gp5/M of PRRSV-2 prototype strain VR 2332. (**A**) Simplified scheme of the structure. Helical regions are drawn as cylinders and β-strands as arrows. Gp5 is colored in green and M in light blue. (**B**) Hydrophobic surface representation of the model. Hydrophilic residues are colored in light blue and hydrophobic residues in yellow–brown. (**C**,**D**) Cartoon model of the structure. The transmembrane parts of Gp5 and M are colored in dark and light grey, respectively, and the ecto- and endodomains of Gp5 and M are colored in green and cyan. Blue spheres: N-glycosylation sites. Orange spheres: palmitoylated cysteine residues. Red sticks in the ectodomain indicate the disulfide bond between Gp5 and M, and D58 is a negatively charged Asp in TM2. A, B and C are shown from the same point of view, and the model in (**D**) is rotated clockwise by 90°. (**A**,**B**) were created with Chimera X, and (**C**,**D**) with Pymol from model 1. Note that the numbering of amino acids starts with the first residue in the mature protein and therefore 31 must be added to obtain the number in the protein including its signal peptide.

**Figure 3 ijms-23-13209-f003:**
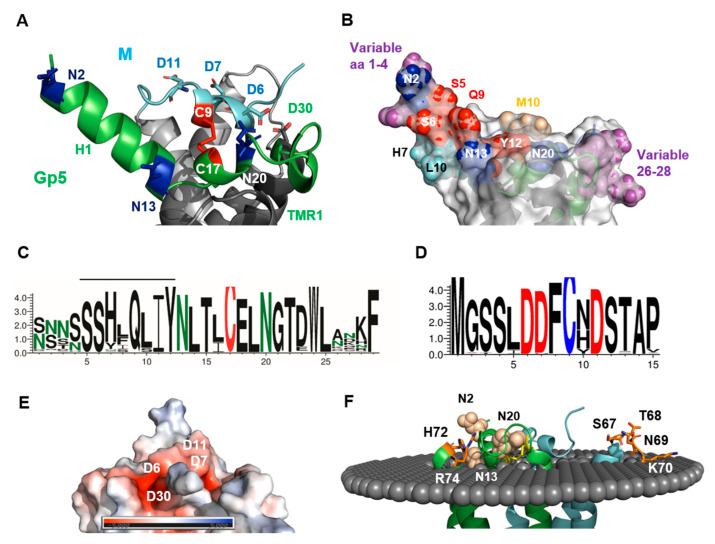
The structure of the ectodomain of Gp5/M of PRRSV-2. (**A**) Cartoon model. Hydrophilic residues in Gp5 are colored green and in M are colored cyan. Hydrophobic membrane parts of Gp5 are colored dark grey and of M are colored light grey. N-glycosylated Asn are shown as blue sticks, residues forming the disulfide bond between Gp5 and M are shown as red sticks and Asp residues in Gp5 and M are shown as green and cyan sticks, respectively. (**B**) Surface representation of the ectodomain. N-glycosylation sites are shown as blue spheres, and between PRRSV-2 strains, variable amino acids are shown as magenta spheres. The neutralizing epitope (residues 5–12) is also highlighted, conserved residues are highlighted as red spheres and the between strains variable residues are highlighted as cyan spheres. Residue 10 of M, which is also involved in antibody binding, is shown as a wheat sphere. (**C**,**D**) Web logo showing the amino acids at each position in the ectodomain of Gp5 (**C**) and M (**D**). Logos were generated from 7701 aligned Gp5 and 290 aligned M amino acid sequences from PRRSV-2 strains. The overall height of the stack indicates the sequence conservation at a position (x-axis), while the height of symbols within the stack indicates the relative frequency of each amino. Asn residues are labeled green, Cys is labeled red (**C**) or blue (**D**) and Asp residues are labeled (**D**) in red. The neutralizing epitope in Gp5 is highlighted by a black line. (**E**) Electrostatic surface potential of the Gp5/M ectodomain. The location of negatively charged Asp residues is indicated. The structure is rotated clockwise by 90° relative to the structure shown in A. The scale bar indicates the electrostatic surface potential from −5 (deep red) to +5 (deep blue). (**F**) Integration of Gp5/M into a virtual lipid bilayer. The border between the hydrophobic and hydrophilic part of the external part of the bilayer is shown as grey spheres. Gp5 and M are shown as green and cyan cartoons, respectively. The N-glycosylation sites are shown as wheat spheres, and amino acids located in the loop between transmembrane helices 2 and 3 of both Gp5 and M and protruding from the membrane are shown as orange sticks.

**Figure 4 ijms-23-13209-f004:**
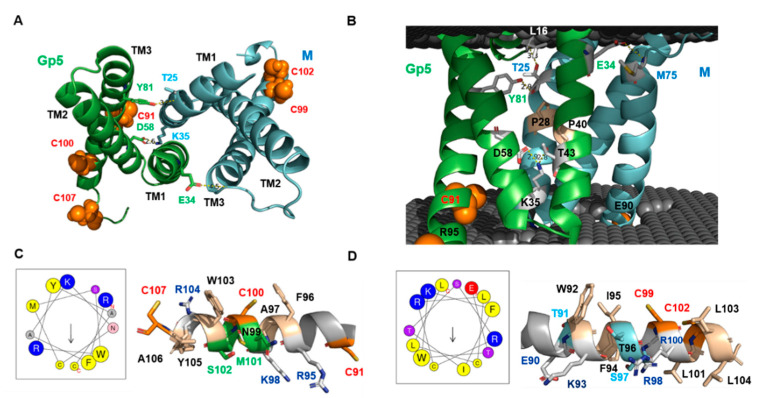
The structure of the transmembrane region of Gp5/M of PRRSV-2. (**A**,**B**) Top view and side view of the transmembrane region, the latter embedded in a virtual lipid bilayer. Palmitoylated Cys in Gp5 and M are shown as orange spheres. Conserved Pro in the middle of TM1 of both Gp5 and M are highlighted as wheat sticks in (**B**). Amino acids that form ionic interactions between Gp5 and M are highlighted as sticks. Asp58 forms an ionic bond with Lys35 in M, which is located in the loop between TM1 and TM2 and also interacts with Thr43 in TM1 of Gp5. In the outer membrane leaflet, the hydroxyl group of Tyr81 in TM3 of Gp5 interacts with the main chain in TM1 of M at position Thr25. The side chain of this Thr interacts with the main chain of the ectodomain of Gp5 at position Leu16. Finally, Glu34 in TM1 of Gp5 interacts with the main chain of M at position 75 located at the end of the loop connecting TM2 and 3. (**C**,**D**) Heliquest analysis of the C-terminal region of TM3 of Gp5 (**C**) and M (**D**) and structure of the corresponding domain. Heliquest produces a helical wheel from the sequence with an arrow inside that points to the hydrophobic face. Its length corresponds to the hydrophobic moment. A hydrophobicity <H> of 0.483 and a hydrophobic moment <µH> of 0.447 was calculated for Gp5, whereas the corresponding values for M are <H> 0.723 and <µH>: 0.335.

**Figure 5 ijms-23-13209-f005:**
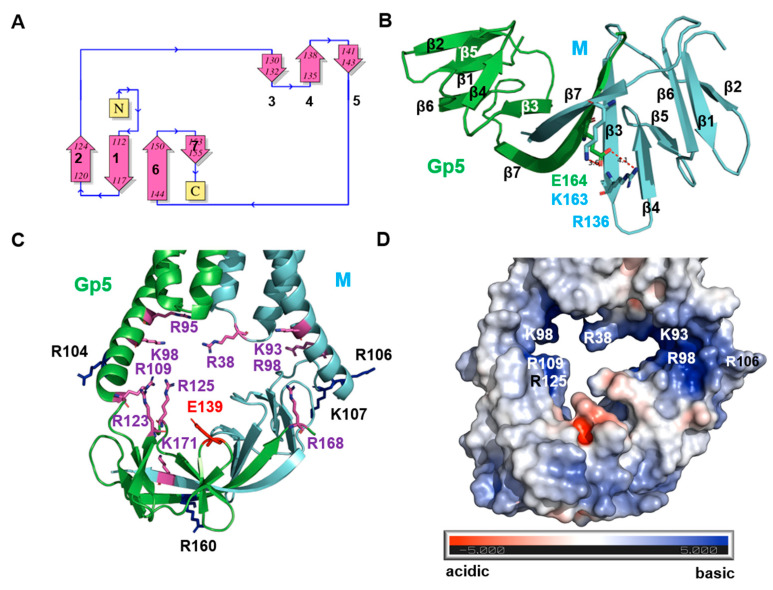
The structure of the endodomain of Gp5/M of PRRSV-2. (**A**) Topology sketch of the endodomain of M. β-strands 1, 2, 6 and parts of 7 build one β-sheet, and β-strands 3, 4 and 5 build another β-sheet. Note that the topology of the endodomain of Gp5 is the same. (**B**) Cartoon model of the endodomain, Gp5 in green and M in cyan. The numbering of the β-strands is indicated. Gp5 and M interact mainly via parts of β-strands 7 and via a salt bridge formed between the carboxyl group of Glu164 in β7 of Gp5 with the amine group of Arg163 located in β7 of M and with the guanidino group of Lys136 located in β3 of M. (**C**,**D**) Cartoon representation (**C**) and electrostatic surface potential of the surface (**D**) of the cytoplasmic part of TM1 to TM3 and of the endodomains of Gp5 and M. Basic amino acids are highlighted as magenta sticks They are located in loop 1 connecting TM1 and 2 of M (Arg38) in the amphiphilic helix (Arg95, Lys98) and in the endodomain of Gp5 (Arg123, Arg125). Lys93 and Arg98 in the amphiphilic helix of M and Arg168 in the endodomain of M form another strongly basic area on the other side of the cavity.

**Figure 6 ijms-23-13209-f006:**
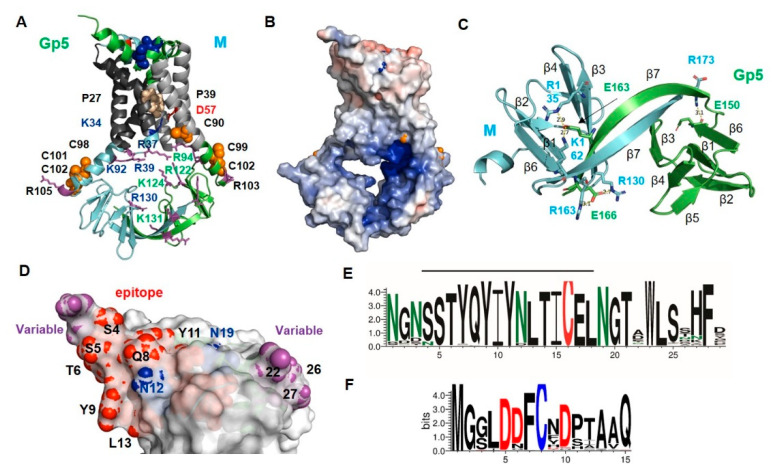
The predicted structure of Gp5/M of PRRSV-1 prototype strain Lelystad. (**A**) Cartoon model. Ecto- and endodomains are colored green in Gp5 and cyan in M. Hydrophobic membrane parts of Gp5 are colored dark grey and of M are colored light grey. N-glycosylated Asn are shown as blue sticks, and the disulfide bond between Gp5 and M are shown as red sticks. Conserved Pro in TM1 of Gp5 and M are shown as wheat sphere. The amino acids forming an electrostatic interaction between TM2 of Gp5 (D57) and TM1 of M (K34) are shown as sticks. Palmitoylated Cys in Gp5 and M are highlighted as orange spheres, and basic amino acids in the endodomain are shown as magenta sticks. See Appendix A for the corresponding quality scores of the model. (**B**) Electrostatic surface potential. The orange spheres are the acylation sites in Gp5 and M. (**C**) Cartoon of the endodomain, Gp5 in green and M in cyan. The numbering of the β-strands is indicated. Gp5 and M interact mainly via parts of β-strands 7 and via three salt bridges. The carboxyl-group of Glu166 in β7 of Gp5 interacts with the guanidino groups of Arg163 (located in β7) and Arg130 (in β3) in M. In a similar manner, Glu163 in β7 of Gp5 interacts with Arg135 (in β3) and with the amine group of Lys162 (located in β7) in M. Finally, Arg 173 at the C-terminus of β7 in M interacts with Glu150 in β6 of Gp5. (**D**) Surface representation of the ectodomain. N-glycosylation sites are shown as blue spheres, and between PRRSV-1 strains, variable amino acids are shown as magenta spheres. Amino acids of the neutralizing epitope (residues 5–18) are highlighted as red spheres. (**E**,**F**) Web logo showing the amino acids at each position in the ectodomain of Gp5 (**E**) and M (**F**). Logos were generated from 1077 aligned Gp5 and 83 aligned M amino acid sequences from PRRSV-1 strains. The overall height of the stack indicates the sequence conservation at a position (x-axis), while the height of symbols within the stack indicates the relative frequency of each amino. Asn residues are labeled green; Cys are labeled red in E and blue in F; and Asp residues in F are labeled red. A graph showing the percent amino acid conservation at each position is shown in Appendix A. The epitope in Gp5 is marked with a black line. Note that the numbering of amino acids starts with the first residue in the mature protein and therefore 34 must be added to obtain the number in the protein, including its signal peptide.

**Figure 7 ijms-23-13209-f007:**
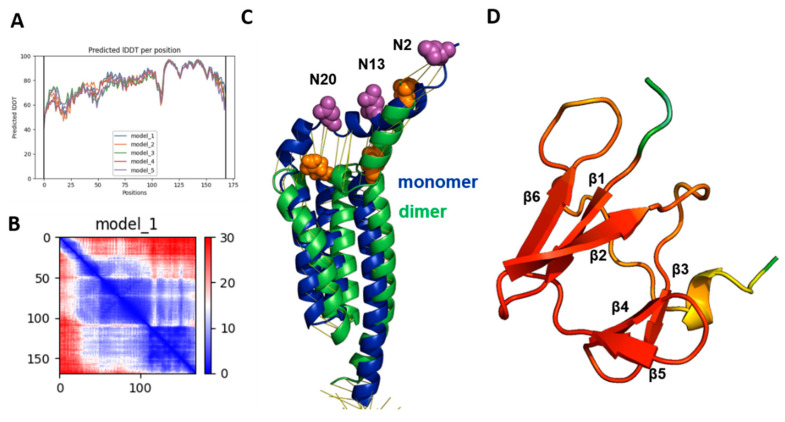
Possible conformational changes occurring upon dimerization of Gp5/M. (**A**) Predicted local distance difference test (LDDT) score per position for the five models generated by alphafold2 for monomeric Gp5 from VR 2332. (**B**) Prediction aligned error (PAE) score for model 1. (**C**) Alignment of the ectodomain and transmembrane region of monomeric Gp5 (blue) with the corresponding region of the Gp5/M dimer (green) from VR 2332. The N-glycosylation sites are highlighted as magenta (monomer) or orange (dimer) spheres. (**D**) Endodomain of Gp5 from VR 2332 showing the pLDDT per position in rainbow colors from red (high confidence) to blue (low confidence) and the numbering of β-sheets.

**Figure 8 ijms-23-13209-f008:**
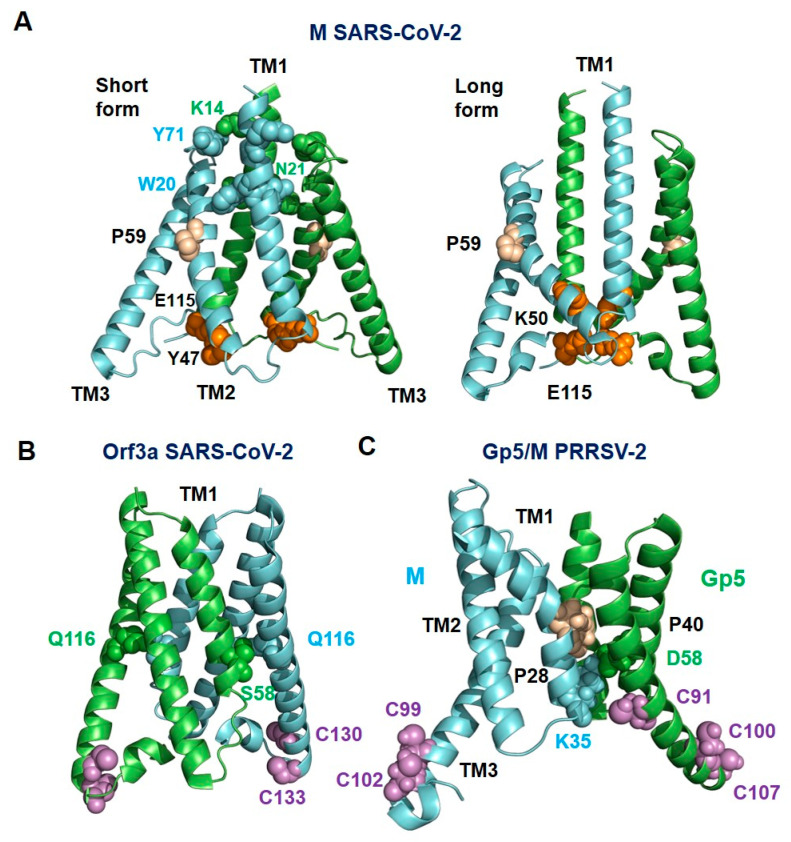
Structure of the transmembrane region of the short and long form of M (**A**), Orf3a (**B**) of SARS-CoV-2 and Gp5/M of PRRSV-2 (**C**). (**A**). One monomer of Orf3a and M of SARS-CoV-2 is colored green, and the other is colored cyan. Gp5 is colored green and M is colored cyan in the predicted structure of Gp5/M of the PRRSV-2 reference strain VR 2332. Proline residues in TM2 of M of SARS-CoV-2 and TM1 of Gp5/M are highlighted as wheat spheres, and cysteine residues at the C-terminus of TM3 of Gp5/M and Orf3a are highlighted as magenta spheres. Residues that form hydrophilic interactions between the monomers are highlighted as spheres with the color of the respective monomer. Residues forming interactions between the hinge region and TM2 in the short and long form of M of SARS-CoV-2 are highlighted as orange spheres. Figures were created with Pymol from PDB-file 7VGS (M, short form), 7VGR (M. long form) and 6XDC (Orf3a).

**Figure 9 ijms-23-13209-f009:**
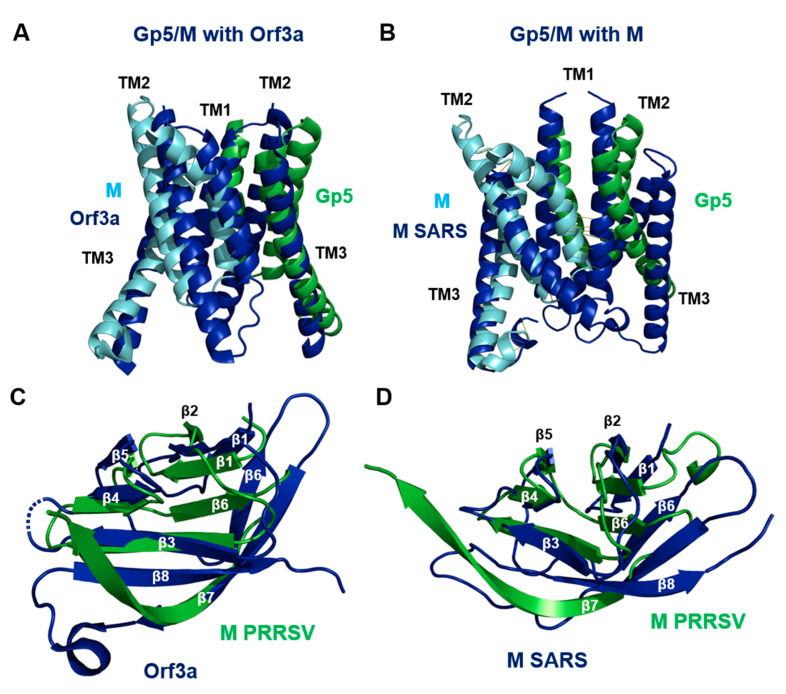
Alignment of the transmembrane regions of Gp5/M of PRRSV-2 with Orf3a (**A**) and with the long form of M (**B**) of SARS-CoV-2. Gp5 is colored green; M is colored cyan; and Orf3a and M of SARS-CoV-2 are colored blue. (**C**,**D**) Alignment of the endodomain of M of PRRSV-2 with one endodomain of Orf3a (**C**) or M (**D**) of SARS-CoV-2. M of PRRSV is colored green, and Orf3a and M of SARS-CoV-2 are colored blue. The transmembrane helices and β-sheets in the endodomain are numbered. Figures were created with Pymol from PDB-file 7VGR (M. long form) and 6XDC (Orf3a).

**Figure 10 ijms-23-13209-f010:**
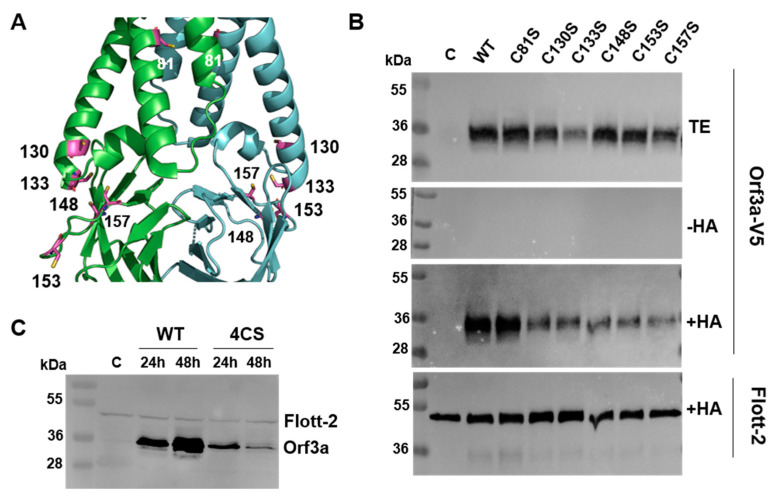
Orf3a of SARS-CoV-2 is palmitoylated at membrane-near cysteine regions in the cytoplasmic tail. (**A**) Cartoon structure of Orf3a with cysteine residues highlighted as magenta sticks. (**B**) Acylation analysis of wt and mutant Orf3a protein. Upper panel: The Orf3a gene (wt and the mutants), where individual cysteines were substituted by serines and were fused at the C-terminus to a V5 tag and expressed in 293T cells. To test for protein expression, 10% of the cell lysate was removed. Middle and lower panel: The remainder of the lysate was divided into two aliquots, one not treated (−HA) and one treated with hydroxylamine (+HA) to cleave cysteine-bound fatty acids before pulling down proteins with a free SH group. Samples were subjected to Western blotting with antibodies against the V5 tag and, subsequently, with those against the endogenous protein flotillin-2 (Flot-2). Molecular weight markers are shown on the left of each blot. (**C**) Stability analysis of Orf3a. Wt Orf3a and a mutant, where the four acylated cysteines 130, 133, 148 and 153 were exchanged to serine, were expressed in 293T cells, which were lysed 24 or 48 h after transfection. Identical aliquots of the cell lysate were subjected to Western blotting with antibodies against the V5 tag and, subsequently, with those against the endogenous protein flotillin-2 (Flott-2) as the loading control. C: untransfected control cells.

## Data Availability

All data are available in the manuscript, including the Appendix A. The PDB files of the models were deposited on 17 October 2022 in ModelArchive (modelarchive.org) with the accession codes ma-a61pn (PRRSV-2 VR 2332, model 1) and ma-pwhoi. (PRRSV-1 Lelystad, model 1).

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
