# Peer review of "Using Alphafold2 to Predict the Structure of the Gp5/M Dimer of Porcine Respiratory and Reproductive Syndrome Virus"

_ijms, 2022, doi:10.3390/ijms232113209_

Round 1

Reviewer 1 Report

The paper on hand uses the novel AI prediction of protein structure using alphafold2 to predict the GP5 / M dimer of PRRSV. Overall, this is an interesting paper to the field and the introduction in particular gives a very good overview over the knowledge on PRRSV. 

However, the paper is much too long and whilst processes are explained in much detail, they are only brought in context in the discussion whilst results already provide lots of context.

The first question I asked myself was what are the interesting parts of this paper or of the work performed and based on this I would suggest a bit of restructure. 

1) comparison of structures between PRRSV-1 and PRRSV-2

2) interrogating individual structures and epitope shielding side-by-side of the two different species

3) highlighting the impact of dimerisation and functional implication of this process. 

4) Whilst generally tangientially interesting, I am not sure whether the orf3/M SARS-CoV-2 comparison adds any value to this paper. Writing in this section is also highly confusing as it is not always clear which M protein (PRRSV or SARS) the authors talk about.  

Specific areas where more attention should be paid to:

1) Structure of the dimer and interaction with partners. 

Whilst the authors discuss epitopes, they do not discuss glycosylation and palmitoylation within the structures. These facts are introduced in the introduction but not sufficiently addressed in the structure prediction and results section. 

The section on the transmembrane domain does highlight and answer previous questions on the GP5 transmembrane helix but it does not address important functions found in other viruses for these transmembrane proteins inducing and mediating membrane curvature (which is eventually discussed in the discussion section). However, the section on dimerisation highlights a change to the helix structure. This must be discussed right in the respective sections to highlight functional implications. 

In the endodomain, Interaction with other protein partners of the virion should be discussed. 

2) Are there differences between PRRSV-1 and PRRSV-2 but also, are there differences between highly divergent subvariants of the virus since as the authors highlight correctly, GP5 is used to classify PRRSV subvariants. 

This question is barely touched on nor are the structures between the two species compared side-by-side. 

3) If the comparison with orf3a of SARS-CoV-2 is included, basic considerations must be made, including highlighting how membrane curvature is induced in SARS-CoV-2 and whether orf3a plays any role in forming the virion. orf3a is a dimer or tetramer and comparisons or highlighting of different structures and potential interactions of these structures must be considered. I personally thing the section on SARS-CoV-2 is tangiential at best and should be improved or removed. Improvements would include a side-by-side comparison and a functional comparison on induction of membrane curvature. Otherwise comparisons would  be discussed in the the discussion section only. 

Reviewer 2 Report

The authors used AlphaFold 2 to predict a structure of Gp5/M. In the abstract, the authors described some structural features of the predicted structures for the Gp5/M complex and proposed mechanisms of virus budding and models of antibody-dependent virus neutralization involving Gp5/M. I do appreciate that the authors have included a detailed description of all the tools/servers used in this manuscript. The choices of computational tools make sense, and most of them have been in the field for a while.

The authors have invested their time and tried to use multiple computational software/servers to explore the Gp5/M system based on the AlphaFold 2 predicted structure models. However, the manuscript must go through a major revision because the authors failed to compose a coherent manuscript based on tons of computational results reported in the manuscript. What is the main message this manuscript aims to convey? Obtaining a computationally predicted structure of Gp5/M? If so, extensive evaluations of the AlphaFold 2 predicted structures should be done using data other than AlphaFold 2 reported matrices. Why should the readers care about the peptide cleavage sites in Gp5 proteins? Should the title and abstract reflect that if the authors aim to explore/interpret how this protein performs its biological functions based on the AlphaFold 2 predicted structures? Computational tools are powerful but should be used carefully.

The following points need to be addressed:

[1]. AlphaFold 2 can predict most of the proteins with reasonable accuracy. Though many research groups started to use AlphaFold 2 to carry out predictions for complex structures, the accuracy is questionable. In the case of Gp5/M, the transmembrane part is probably reliable, but the bottom beta sheets region would be challenging to predict, especially when the authors did not use templates. Could the authors rerun your AlphaFold predictions with templates to explore the differences in the predicted structures? Also, I am not sure three recycles would be sufficient for the predictions to converge. The author should check some of the recent AlphaFold publications (Curr. Opin. Struct. Biol., 74:102372, 2022; Biomolecules, 12(7), 985, 2022) besides the ones cited in the manuscript.

[2]. The AlphaFold database (https://alphafold.ebi.ac.uk/) has recently been updated. It is important to check against the homologs of Gp5/M reported in the database to support the predicted structures by the authors, in particular, for sections 2.9 and 2.10. 

[3]. I have no idea what the authors were talking about in section 2.2. What are Figures 2C and 2D for? Please also note that the figure legend is missing a description for panel C. Only Figure 2B was cited.

[4]. It is great to have all these computationally predicted results. The authors must compare their results with previously reported experimental/computational results.  

[5]. I do not know why the authors did not carry out experiments for Gp5/M which is the focus of this manuscript. Though Orf3a and Gp5/M are similar, the manuscript is for Gp5/M. 

Round 2

Reviewer 2 Report

The revision looks good to me. However, one sentence must be removed before the acceptance of the manuscript. 

The beta region, as shown in Fig. 1B, clearly consists of a low-confidence region. In the youtube video, John stated that a high confidence score would be above 90. The rank 1 model is good if pLDDT is considered, terrible in PAE in terms of internal packing of the complex. Rank 2 is better in PAE, but a significant portion of the sequence has pLDDT lower than 70. Therefore, the newly added sentence "this is the first reliable structure of an oligomeric viral protein predicted de novo with alphafold2" must be deleted. It is necessary to point out that AlphaFold 2 failed to predict some targets in CASP14.  

Author Response

Reply to referee 2:

The revision looks good to me. However, one sentence must be removed before the acceptance of the manuscript.

The beta region, as shown in Fig. 1B, clearly consists of a low-confidence region. In the youtube video, John stated that a high confidence score would be above 90. The rank 1 model is good if pLDDT is considered, terrible in PAE in terms of internal packing of the complex. Rank 2 is better in PAE, but a significant portion of the sequence has pLDDT lower than 70. Therefore, the newly added sentence "this is the first reliable structure of an oligomeric viral protein predicted de novo with alphafold2" must be deleted. It is necessary to point out that AlphaFold 2 failed to predict some targets in CASP14. 

We agree with the opinion of the referee. We have modified the manuscript as follows, also noting that alphafold2 cannot deliver a trustworthy prediction for every protein and that the confidence scores are crucial to assess the model´s quality:

“… this is the first in many parts reliable structure of an oligomeric viral protein predicted de novo with alphafold2. However, alphafold2 cannot yet successfully predict every viral protein, as we have not been able to obtain a reliable structure for the membrane proteins Gp2, Gp3, and Gp4 of PRRSV. The confidence values are crucial to assess whether the predicted models (and which parts of it) are reliable.